# The Hyperfitting Phenomenon: Sharpening and Stabilizing LLMs for Open-Ended Text Generation

**Fredrik Carlsson**[*]    **Fangyu Liu**[†]    **Daniel Ward**[§]    **Murathan Kurfali**[*]    **Joakim Nivre**[‡]
[*]RISE Research Institutes of Sweden  [†]Google DeepMind  [§]PwC Sweden  [‡]Uppsala University

## Abstract

This paper introduces the counter-intuitive generalization results of overfitting pre-trained large language models (LLMs) on very small datasets. In the setting of open-ended text generation, it is well-documented that LLMs tend to generate repetitive and dull sequences, a phenomenon that is especially apparent when generating using greedy decoding. This issue persists even with state-of-the-art LLMs containing billions of parameters, trained via next-token prediction on large datasets. We find that by further fine-tuning these models to achieve a near-zero training loss on a small set of samples – a process we refer to as hyperfitting – the long-sequence generative capabilities are greatly enhanced. Greedy decoding with these Hyperfitted models even outperforms Top-P sampling over long-sequences, both in terms of diversity and human preferences. This phenomenon extends to LLMs of various sizes, different domains, and even autoregressive image generation. We further find this phenomena to be distinctly different from that of Grokking and double descent. Surprisingly, our experiments indicate that hyperfitted models rarely fall into repeating sequences they were trained on, and even explicitly blocking these sequences results in high-quality output. All hyperfitted models produce extremely low-entropy predictions, often allocating nearly all probability to a single token.

## 1 Introduction

Despite the recent rapid advancements in artificial intelligence spearheaded by Transformer-based large language models (LLMs) and their emergent phenomena (Wei et al., 2022b; Bubeck et al., 2023), models trained on next-token pre-training objectives often degenerate when producing longer texts. This is particularly true for greedy decoding, and has resulted in mitigation strategies such as repetition penalties (Keskar et al., 2019) and nucleus sampling (Holtzman et al., 2020). However, when removing these heuristics and simply picking the top-1 candidate at each time-step, LLMs display strong tendencies to repeat themselves at the token, phrase, and sentence level (Holtzman et al., 2020), as is exemplified in Figure 1. This is a recurrent phenomenon for which there are many proposed hypotheses but, to the best of our knowledge, no definitive explanation exists.

**Context**

Yardley's batting form dipped in the 1950 season. He scored 1,082 runs at an average

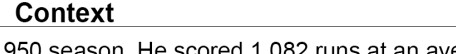

| | |
|---|---|
| of 29.90 with a top score of 108. With the ball, he had a figure of 1,099 runs at an average of 30.90 with a best of 109. He was named in the team for the first time in 1950. He played in all the three tests and the two one day internationals that were played that year. He made his test debut against the ... | of 26.95, with a highest score of 100 not out. He played in 41 first-class matches, scoring 1,082 runs at an average of 26.95, with a highest score of 100 not out. He played in 41 first-class matches, scoring 1,082 runs at an average of 26.95, with a highest score of 100 not out. He played in 41 first-class matches ... |
| *Hyperfitted* **LLama 3.1 (8B)** | **LLama 3.1 (8B)** |

Figure 1: Example of greedy decoding using Llama 3.1 and its hyperfitted counterpart. Color indicating how repetitive the generated text is.

In this paper, we report on the counter-intuitive discovery that overfitting a pre-trained LLM on a very small set of samples until it achieves a near-zero training loss – a process we refer to as hyperfitting – greatly enhances the greedy decoding capabilities. Although these models achieve significantly worse validation loss, they produce texts that align markedly better with human preferences and automatic diversity metrics. Indeed, we find that hyperfitting state-of-the-art LLMs yields capabilities that outperform models with 10x the number of parameters. Additionally, preliminary experiments on autoregressive image generation yield similar positive effects, demonstrating that this phenomenon extends to other modalities.

Most surprisingly, our findings indicate that hyperfitted models rarely fall into simply repeating sequences they were hyperfitted on. Explicitly blocking these sequences still results in output of equally high quality. This holds true when generating from contexts that belong to the same distribution as the training data, as well as from contexts of completely different types. Notably, hyperfitted models produce a sharpened modelling space, predicting distributions with significantly lower entropy that often favors a single candidate token at each time step.

Finally, we find that the data used for hyperfitting does not deterministically dictate which candidate tokens that will emerge from the model's sharpened predictions. Rather, the narrowing of the modeling space is also a product of the training process itself, as hyperfitting on identical data, but shuffled, results in noticeably different predictions. Hence, we hypothesize that the improvement in long-sequence generation is due to the collapse and sharpening of the corpus-average modeling space attained during pre-training. Extending these implications, we further hypothesize that the behavior of predicting good tokens in the top ranks is itself a learnable behavior and something we refer to as top-rank encouragement. Skeleton code and models available at: github.com/FreddeFrallan/Hyperfitting

## 2 RELATED WORK

Various recent studies report phenomena regarding neural networks that break the conventional practice of early stopping. For instance, "double descent" shows a second rise and decline in test loss beyond the classical bias-variance trade-off Belkin et al. (2019); Nakkiran et al. (2020). Overfitting networks for a prolonged duration may lead to "grokking", resulting in strong delayed generalization Power et al. (2022); Liu et al. (2023). There are recorded cases of benign overfitting, where a model that perfectly fits noisy training data still generalizes well to unseen scenarios Zhang et al. (2017); Belkin et al. (2019). These phenomena, which seemingly break the foundations of statistical learning theory, are often attributed to the over-parameterization of large networks in relation to the training data volume He et al. (2016); Zhang et al. (2021).

It is well documented that in long-sequence text generation LLMs exhibit degenerative tendencies such as repetition (Welleck et al., 2019; Holtzman et al., 2020; Fu et al., 2020; Brown et al., 2020; Xu et al., 2022). While mitigation strategies like repetition penalties (Keskar et al., 2019) and sophisticated sampling strategies (Holtzman et al., 2020) exist, no definitive explanation for why this occurs has been given. Interestingly, repetitive texts tend to occur less frequently for conditional generation tasks, such as machine translation and summarization (Holtzman et al., 2020).

Although next-token prediction is the dominant training objective for LLMs, it is not perfectly aligned with the requirements of sequence generation (Welleck et al., 2019; Bachmann & Nagarajan, 2024). This is evident in practical scenarios, where alternative approaches may score higher on human preference but achieve lower perplexity (Carlsson et al., 2024). Even in situations of pure language modeling, the next-token prediction loss and its exponentiated version, perplexity, only capture a subset of the statistical properties of language (Meister & Cotterell, 2021).

Our work also relates to the dominant LLM paradigm of pre-training on large datasets, followed by additional fine-tuning in which scaling up next-token prediction training gives rise to emergent and unpredictable capabilities (Wei et al., 2022a; Srivastava et al., 2023). Often, further adjustments to the LLM are made using reinforcement learning for instruction-following (Ouyang et al., 2022). While the main idea of this may be to modify the LLM's interaction API, it can significantly increase long-sequence generation capabilities, as consistently seen in code generation (Guo et al., 2024; Lozhkov et al., 2024). Contrary to these methods, hyperfitting allows us to observe the effects of collapsing the modeling space attained during pre-training, without incorporating any new data or training methods.

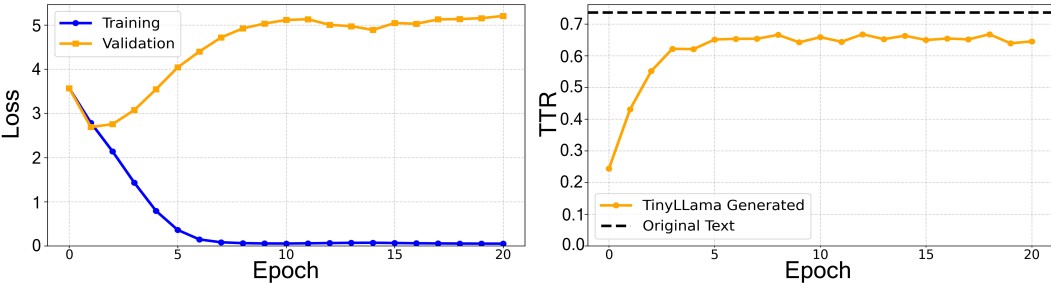

Figure 2: Training and validation loss for TinyLlama during overfitting, along with the resulting mean TTR when greedily generating 96 tokens from contexts in the validation data.

## 3 HYPERFITTING

Hyperfitting is conceptually straightforward and consists of fine-tuning a pre-trained model on a small set of samples until the model achieves a near-zero training loss. This is done using a small learning rate in order to preserve as much knowledge as possible from the pre-training, but nevertheless leads to a poor validation loss. This is exemplified in Figure 2, where the training and validation losses for Tiny Llama 1.1b (Zhang et al., 2024) is shown next to its type-token ratio (TTR) for sequences generated on held-out contexts. TTR is a simple metric measuring the ratio of unique tokens, defined as TTR = $\frac{\text{Number of Unique Tokens (Types)}}{\text{Total Number of Tokens}}$. Although a high TTR does not guarantee textual quality, the average TTR has been shown to correlate well with human preferences for long-sequence text generation (Carlsson et al., 2024), and is further discussed in Appendix A.1s.

To verify that hyperfitting has a reproducible effect, we perform this training on various model instances, datasets and modalities. Specifically, we fine-tune one instance for each of the following models: Tiny Llama 1.1b, DeepSeek 7b (Bi et al., 2024), Llama 3.1 8b & 70B (Dubey et al., 2024), and ImageGPT-Large (Chen et al., 2020) for image generation. Notably, the text models cover a range of quality levels; Llama 3.1 and DeepSeek are, at the time of writing, considered state of the art for open-source LLMs, while TinyLlama is less competitive.

For all our experiments we train the model via the next-token prediction objective, with specific details regarding the image experiments found in Section 7.1. Unless otherwise specified, all LLMs use the following training setup: 20 epochs on 2000 randomly selected sequences from a given dataset, with a length of 256 tokens. We update all the model's parameters using the Adam optimizer with a learning rate of 1e-6 without weight decay, and use a batch size of 8. All hyperfitted LLMs use the identical samples from the Fiction-Stories dataset (Forsythe, 2024). However, as demonstrated in Section 6.2 and Section 7.1, hyperfitting occurs for various datasets and modalities.

Due to the obvious concern of a hyperfitted model only repeating the data it has been fine-tuned on, we additionally generate texts using a citation blocker. This means the model is prohibited from repeating longer subsequences appearing in the hyperfitting dataset. Via a straightforward pattern matching approach, we continuously check if the 5 most recently generated tokens exist as a sequence in the training data. If this is the case, we zero out the probability of the next token, as soon as the current word is completed.

## 4 OPEN-ENDED TEXT GENERATION

To thoroughly evaluate the models' ability to generate text in an open-ended setting, we conduct an extensive human evaluation study with verified English speakers independently hired as freelancers.[1] Each sample consists of a textual context and two possible continuations, with one continuation always coming from the original text and the other generated by a model. The annotator's task is to determine the preferred continuation, or classify them as equally good options. Further details about the annotations are available in Appendix A.

---

[1]https://fiverr.com/

Table 1: Comparison of perplexity over contexts, human preference of generated texts, and lexical variation (as measured by TTR). All models use greedy decoding besides *Llama 3.1 (8 B) Top-P*.

| Model | Context PPL | 128 Pref | 256 Pref | 128 TTR | 256 TTR |
|---|---|---|---|---|---|
| Original Texts | – | – | – | 73.5 | 73.8 |
| **Strong Baselines** | | | | | |
| TinyLLama (1.1 B) Top-P | 245 | 31.8 | 21.1 | 38.8 | 28.2 |
| DeepSeek (7 B) Top-P | 34 | 50.0 | 35.6 | 58.2 | 49.7 |
| Llama 3.1 (8 B) Top-P | 36 | 50.5 | 38.5 | 62.1 | 57.0 |
| **Original Models** | | | | | |
| TinyLlama (1.1 B) | 245 | 12.0 | 4.9 | 25.1 | 17.0 |
| DeepSeek (7 B) | 34 | 37.7 | 17.1 | 45.6 | 32.2 |
| Llama 3.1 (8 B) | 36 | 35.0 | 25.6 | 48.5 | 34.5 |
| Llama 3.1 (70 B) | **29** | 48.7 | 34.4 | 56.4 | 50.6 |
| **Hyperfitted Models** | | | | | |
| TinyLLama (1.1 B) | 467 | 44.6 | 34.3 | 64.5 | 60.0 |
| DeepSeek (7 B) | 545 | 49.4 | 45.2 | 62.3 | 60.5 |
| Llama 3.1 (8 B) | 389 | 50.1 | 42.9 | 64.5 | 62.6 |
| Llama 3.1 (70 B) | 255 | **55.9** | **52.4** | 62.0 | 61.6 |
| **Hyperfitted Models + Citation Blocking** | | | | | |
| TinyLLama (1.1 B) | 467 | 45.2 | 35.0 | 64.8 | 60.3 |
| DeepSeek (7 B) | 545 | 47.5 | 44.1 | 62.5 | 60.6 |
| Llama 3.1 (8 B) | 389 | 47.6 | 41.2 | 64.4 | 63.3 |

Each model generates 100 continuations for each of three datasets: Wikipedia (Merity et al., 2017), Fictional Stories (Forsythe, 2024), and BBC News (Li et al., 2024). These 300 texts are manually validated to have high quality and trimmed to 256 tokens. Of these, we select the first 32 tokens as context and keep the remaining 224 as the original continuation. Additionally, we create a 128-token scenario, where the model's first 96 tokens are compared to the first 96 tokens from the original continuation. For both length scenarios, we gather 3 annotations per comparison. All models generate using greedy decoding, besides the strong baselines that uses nucleus sampling with $TopP = 0.9$, $Temp = 0.7$ and $TopK = 50$. Examples of generated texts are available in Appendix C.2.

In Table 1 we report the percentage of times that a continuation was either preferred or judged equally good to the original. We also report the models' perplexity on the 32-token contexts and the TTR of the tokens in the continuation sequences. Since TTR is sensitive to length, we calculate this metric using the last 96 generated tokens for both the 128 and 256 token scenarios.

## 4.1 TEXT GENERATION RESULTS

Hyperfitting drastically increases the human preference ratio. This is particularly true for the 256-token scenario, where the initially worst performing TinyLlama increases from 4.9% to 34.4%, putting it on par with Llama 3.1 70b. While all models perform worse on the 256 scenario, the drop is less drastic for hyperfitted models. Indeed, greedy decoding with hyperfitted models produce both higher human ratings, and a higher ttr when compared with their nucleus sampling counterparts—a method proposed specifically to mitigate repetitions. Citation blocked models see no noticeable drop in performance.

There is a clear correlation with the average TTR and human preference, as the hyperfitted models demonstrate a comparably small drop in both metrics as sequence length increases. Interestingly, all hyperfitted models perform abysmally in terms of perplexity, corroborating the lack of correlation between this metric and a model's ability to generate longer. This is discussed further in Section 5.

## 4.2 DIVERSITY AND DATASET OVERLAP

As hyperfitting is by definition overfitting a model on a tiny set of samples, we investigate how this impacts the model's diversity and overlap with the training dataset. For diversity between generated sequences we apply Self-BLEU, which measures the highest pairwise BLEU score for all pairwise

Table 2: Diversity metrics for generated texts of 96 tokens. Self-BLEU measures diversity within the generated set, while BLEU and Overlap are measured against the dataset. BLEU captures the highest score for any sequence of similar length, and Overlap denotes the longest matching sequence.

| Model | Self-BLEU Avg | Self-BLEU Max | Dataset BLEU Avg | Dataset BLEU Max | Dataset Overlap Avg | Dataset Overlap Max |
|---|---|---|---|---|---|---|
| **Original Models** | | | | | | |
| TinyLLama | 1.5 | 96.8 | 2.0 | 10.0 | 4.2 | 9.0 |
| DeepSeek | 2.4 | 41.2 | 2.6 | 11.9 | 4.5 | 9.0 |
| Llama 3.1 | 0.9 | 33.7 | 1.9 | 5.9 | 4.0 | 7.0 |
| **Hyperfitted Models** | | | | | | |
| TinyLlama | 1.1 | 36.0 | 3.3 | 9.8 | 5.5 | 15.0 |
| DeepSeek | 1.4 | 26.2 | 3.9 | 18.6 | 5.7 | 40.0 |
| Llama 3.1 | 1.0 | 13.5 | 3.6 | 32.1 | 5.8 | 37.0 |
| **Hyperfitted Models + Citation Blocking** | | | | | | |
| TinyLlama | 1.1 | 31.4 | 3.2 | 9.8 | 4.9 | 8.0 |
| DeepSeek | 1.4 | 26.9 | 3.2 | 9.7 | 4.9 | 8.0 |
| Llama 3.1 | 1.0 | 14.7 | 3.0 | 8.5 | 4.7 | 8.0 |

generated sequences; Dataset BLEU represents the maximum BLEU score between a generated sequence and any 96 token subsequence from the dataset; Dataset Overlap measures the longest overlapping subsequence between each generated sequence and the dataset.

For each metric we first calculate the highest score for each generated sequence separately, Table 2 then presents both the average and the maximum of these highest values[2]. In terms of Self-BLEU, the hyperfitted models, both with and without citation blocking, produce texts that are more diverse from one another than texts produced by the original models. As the original models are prone to repetitions, this may indicate that certain repetitive behaviors appear in multiple generated texts.

The distribution of longest overlaps are visualized in Figure 3, for an additional 1000 generated texts per model. From outliers in these distributions, and max values for Dataset BLEU, it is clear that the hyperfitted models occasionally fall back to re-citing from the training data if not blocked. However, considering that the majority of text overlaps are considerably shorter, such occurrences are rare. Indeed, less than 2% of the texts generated without blocking contain overlaps longer than 10 tokens, and the vast majority of overlaps fall in a range similar to that of the original models.

Considering that the average highest BLEU overlap is only a single point higher than the original models, an overwhelming majority of the generated texts do not simply repeat material from the training data. We therefore conclude that hyperfitting causes a generalizable increase in text generation capabilities. More details about the overlapping sequences are available in Appendix B.2. These experiments show that the hyperfitted TinyLLama is citation blocked more frequently compared to its larger counterpart, and that all models are more prone to repeating the dataset when generating from contexts in the fiction datasets.

---

[2]By calculating the highest value followed by the mean and max we attain more insightful and interpretable numbers. If one instead calculates the average directly this leads to extremely low numbers for all metrics.

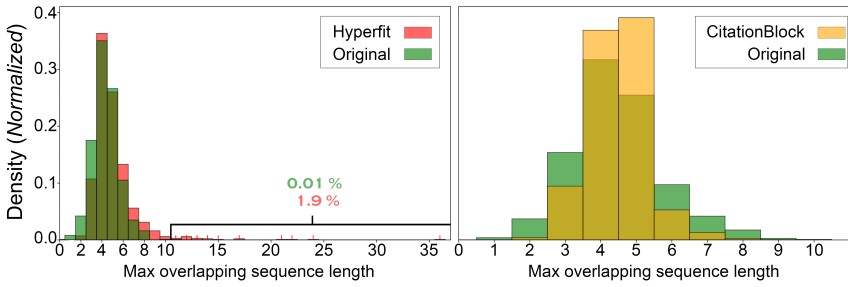

Figure 3: Distribution of the longest overlap between 1000 generated texts and the dataset

Figure 4: A subsequence from the validation data and the corresponding top-3 predictions. The words: "Coverage", "Manchester" and "United" never appear in the hyperfitting dataset.

## 5   SHARPENED PREDICTIONS

Given the boost in textual quality brought about by hyperfitting, we investigate why hyperfitted models achieve such poor perplexity on held-out data. Using the 300 texts and their original continuations from the experiment reported in Section 4, we collect information on the predicted vocabulary distributions of different models. As we observe similar trends across all our hyperfitted models, the results in Table 3 display information only on a subset of the models.

The hyperfitted models exhibit significantly lower entropy in their predicted vocabulary distributions compared to the non-hyperfitted models. This entails that almost all of the probability mass is attributed to a single token. Given the poor perplexity on withheld texts, this sharpened prediction behavior persists even when these predictions are wrong. This persistence is exemplified in Figure 4, where the hyperfitted model assigns "United" a 92.8% probability, although it assigned the previous word "Manchester" a near 0 probability. Neither of these words occur in the hyperfitting dataset.

Hyperfitted models produce vocabulary predictions with extremely low entropy. Moreover, the low training loss indicates that almost all of the probability is consistently assigned to the correct next token during training. This sharpened prediction pattern is, to a degree, transferred to unseen data where the model continues to heavily favor certain candidates. When evaluating these predictions against the unseen data, the low-entropy predictions assign very low probability to words that occur in the new sequences but are not favored by the model, which in turn results in very high perplexity regardless of the quality of the texts they generate. It is worth noting here that, although we follow standard practice and report performance on held-out data using the exponentiated perplexity metric, the key point is really that predictions with low inherent entropy result in high cross-entropy when measured against unseen sequences.

Table 3: Distributions predicted for the original texts (context + continuation) in Section 4. @N indicating the accumulated probability of the N:th tokens with the highest probability.

| Model | Perplexity | Entropy | @1 Prob | @3 Prob | @5 Prob |
|---|---|---|---|---|---|
| **Original Models** | | | | | |
| DeepSeek (7B) | 12.7 | 3.48 | 49.1 | 67.3 | 73.9 |
| Llama 3.1 (8B) | 13.1 | 3.47 | 48.4 | 67.1 | 73.9 |
| Llama 3.1 (70B) | 9.7 | 2.84 | 56.2 | 73.7 | 79.5 |
| **Hyperfitted Models** | | | | | |
| DeepSeek (7B) | 94.7 | 1.32 | 74.5 | 90.0 | 93.3 |
| Llama 3.1 (8B) | 103 | 1.46 | 74.4 | 89.2 | 92.3 |

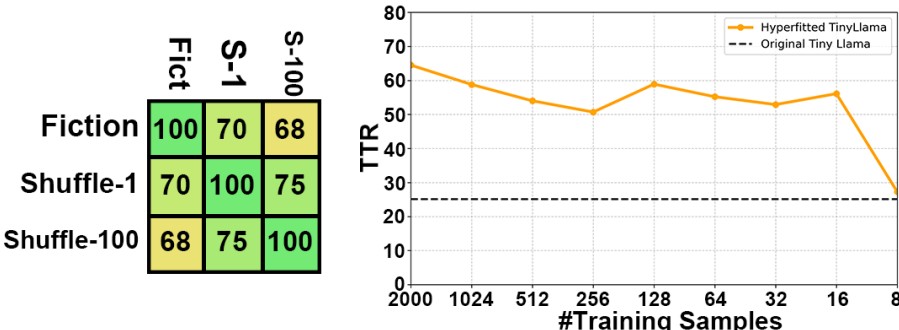

Figure 5: **Left**: Top-1 rank similarity matrix of Llama 3.1 (8B) hyperfitted on identical, but shuffled, data. **Right**: The resulting mean TTR of 300 generated texts as the number of training samples vary.

## 6  DATA INFLUENCE

The following experiments aim to investigate the effect and importance of the data used during hyperfitting. For this endeavor we alter only the data used during hyperfitting and, unless stated otherwise, apply the same training procedure as Section 3. These experiments focus on a single property at a time and do not account for potential relationships between these properties.

### 6.1  DETERMINACY OF DATA

As an initial experiment, we evaluate the extent to which the set of training samples deterministically dictates the outcome of the hyperfitting process. To this end, we produce two additional versions of the fiction dataset: 'Shuffle-1' and 'Shuffle-All'. For Shuffle-1 the order of only two samples is switched, and for Shuffle-All dataset, the entire order is shuffled. For both datasets we hyperfit Llama 3.1 using the same fixed random seed as used in Section 3, meaning all models train on the same data, but in a different order.

Using the full original texts from Section 3, we calculate how often two models produce the same top-1 prediction. This is displayed in the left similarity matrix of Figure 5. All models differ in approximately 30% of their top rank predictions; this is a noticeably large difference from training on the same data. This is especially noteworthy considering that some portion of these predictions will be for subwords, which are almost guaranteed to have the same top rank. Conclusively, the data does not deterministically account for which tokens emerge as top candidates from the hyperfitting process.

### 6.2  TYPE OF DATA

Although Section 6.1 shows that data does not fully determine the resulting model, we nevertheless explore whether any trends emerge between the hyperfitting datasets and downstream dataset capabilities. Therefore, we additionally hyperfit Llama 3.1 with data from both Wikipedia and BBC News separately and measure per-dataset human preference for the 256-token task in Section 3. Qualitative examples of these models are available in Appendix C.2.

Table 4: Human success ratio for 256 token lengths across Fiction, Wiki, and News datasets.

| Model | Fiction Pref | Wikipedia Pref | News Pref | Average |
|---|---|---|---|---|
| **Original Models** | | | | |
| Llama 3.1 (8B) | 21.9 | 26.0 | 24.8 | 24.23 |
| **Hyperfitted Models** | | | | |
| Llama 3.1 (8B) **Fiction** | 41.0 | 40.4 | 40.8 | 40.73 |
| Llama 3.1 (8B) **News** | 59.3 | 77.2 | 62.6 | 66.37 |
| Llama 3.1 (8B) **Wiki** | 55.5 | 52.6 | 44.5 | 50.87 |

The results in Table 4 show that the difference in overall performance between these models is drastic. The news model performs best across all datasets, followed by the Wikipedia model. All of our hyperfitted models consistently outperform their original counterparts. However, no clear trend emerges between the types of training data and the performance on specific datasets. When factoring in the results of Section 6.1, we cannot draw any further conclusions regarding the type of training data and downstream capabilities.

### 6.3 QUANTITY OF DATA

Finally, we measure the effect of the number of training samples from the Fiction dataset when hyperfitting TinyLlama. To this end we keep the number of updates constant at 5000, entailing more epochs as the number of samples decreases. The right part of Figure 5 displays the resulting TTR of the first 96 tokens when greedily generating from the 300 contexts used in Section 4. Since human annotations are costly, TTR is intended as a crude estimate of quality by measuring the repetitiveness of generations. Further discussion regarding TTR as an automatic metric is available in Appendix A.1.

Although there is an initial decline in TTR when decreasing from 2000 samples, the TTR remains above 50 up until there are only 8 training samples. This means that in terms of producing less repetitive output, improvements may be seen from very few samples. Further, we note that 8 samples equals our hyperfitting batch size, meaning that at this point all batch updates are identical, and may hence be indicative of why this is drastically worse than 16 samples.

## 7 HYPERFITTING AND THE BIGGER PICTURE

### 7.1 IMAGE GENERATION

To investigate the hyperfitting phenomenon for an additional modality, we hyperfit ImageGPT-Large (774M parameters) (Chen et al., 2020) on 2,000 randomly selected images from CIFAR-10. Besides using visual tokens, ImageGPT is a standard Transformer architecture and was pre-trained using next-token prediction on 32x32 images. Figure 6 contains a qualitative comparison of the greedy generation when the models receive the first 25% of an image. More details and results are available in Appendix B.4.

From visual inspection it is clear that the hyperfitted model produces higher quality images that more resemble actual objects and subjects (see Appendix B.4 for more examples). Although the generated image quality is unimpressive compared to contemporary diffusion based models, the relative improvement allows us to conclude that the hyperfitting phenomenon extends to other modalities beyond just text. Moreover, we note that greedily generating with ImageGPT results in repetitive patterns analogous to the repetitive texts of LLMs. This strongly indicates that the repetitive nature of Transformer LLMs is not an artifact of the repetitions found in natural language, as posed by Fu et al. (2020) and Holtzman et al. (2020).

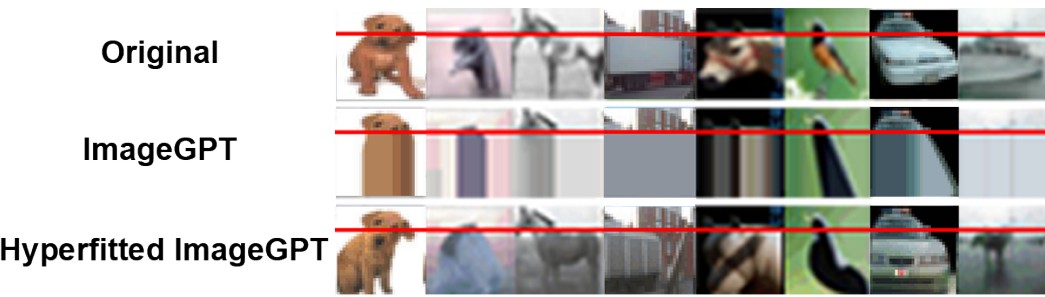

Figure 6: Examples of generated images using greedy decoding when passed 25% of an image.

## 7.2 RELATIONSHIP TO GROKKING AND DOUBLE DESCENT

The hyperfitting phenomenon differs in several key ways from the reported work on grokking and double descent. **(1)** As seen in Figure 2, the positive effect of hyperfitting occurs as training loss approaches zero. In contrast, previous phenomena occur during prolonged exposure to low training loss. **(2)** All our hyperfitted models are pre-trained LLMs with billions of parameters, whereas previous work utilizes comparably small and randomly initialized networks. **(3)** Hyperfitting is observed in the task of sequence generation, where the model's predictions are recursively added to its input. Previously observed phenomena have focused on single output tasks, such as classification and regression. **(4)** Hyperfitting sees improvements in terms of TTR after only a few epochs, as evident in Figure 2, significantly faster than the reported occurrence of double descent and the delayed rewards of grokking. **(5)** None of the hyperfitting training utilizes any form of weight decay, which is speculated to be a main contributor to delayed generalization in grokking (Liu et al., 2023).

From **(2)** and **(3)**, we conclude that hyperfitting is observed at a higher level of model and task complexity. One may argue that if grokking were to occur in large pre-trained models, it would happen quickly and would therefore reconcile **(1)** and **(4)**. However, this is yet to be achieved, and it is unclear if such speed would even be compatible with the slow progress of weight-decay **(5)**. Therefore, besides all phenomena seemingly contradicting early stopping, we currently find no evidence of a commonality. We therefore argue for treating hyperfitting as a separate phenomenon.

Finally, one may note that the validation loss for hyperfitting never decreases, distinguishing it from the hallmark of double descent and grokking. However, as the next-token prediction loss is not fully aligned with our task of sequence generation, we do not consider this to be a reasonable comparison. Admittedly, this entails we cannot track an aligned validation score, preventing us from proving that hyperfitting fundamentally differs from previous discoveries.

## 7.3 TOP-RANK ENCOURAGEMENT HYPOTHESIS

This subsection explores our observation that scenarios with low training loss cause desirable tokens to be ranked higher even when validation loss is poor. For this, we use the term 'top-ranks' to refer to the set of most probable tokens in a predicted distribution, and perplexity as the exponential of the log loss for the next token.[3] The notion of a "desirable" token entails that, if generated, the token would extend the current sequence in a manner acceptable by a human.

Since next-token prediction does not factor in the order and rank of predictions, we note that a higher loss leaves more room for undesired candidates to reside within the predicted top-ranks. In a scenario of moderate perplexity, this means that two models achieving identical perplexity on all time-steps, can still have different top-ranks. This notion is visualized in Figure 7.

A low training perplexity entails distributions during training with low entropy. As observed in Section 5, this entropy behaviour is transferred to validation data. But simply lowering entropy does not by itself entail that top-rank predictions improve. Indeed, we can freely modify the entropy of any (non-uniform) distribution by applying temperature to it, without any change in the predicted order. It follows that something additional happens to the model as it achieves a low training loss.

---

[3]Note again that the important point is not whether the loss is measured on an exponential scale or not, but that it is the cross-entropy with respect to an external sequence, as opposed to the inherent entropy of the predicted probability distribution.

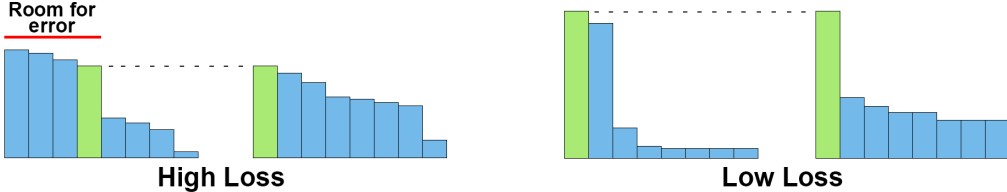

Figure 7: Visualization of how distributions with the same probability for the next token (visualized in green) leave more or less room for error in the top candidates, depending on their entropy.

We hypothesize that training scenarios where a model achieves a low loss teaches the model to prioritize desirable top-rank candidate. We refer to this as top-rank encouragement. Having a desirable token in the top-rank is distinctly different from perplexity, which measures the average probability of the next token over a set of sequences. Section 4 further demonstrates that a model can predict desirable top-rank tokens, despite poor perplexity on the context.

## 8 DISCUSSION AND CONCLUSIONS

We introduce the hyperfitting phenomenon: where pre-trained LLMs consistently see significant improvements in open-ended text generation by overfitting to a very small set of samples. The textual quality of the models are assessed via human verified English speakers, resulting in a new dataset with over 20,000 annotations. We provide extensive evidence that the hyperfitting phenomenon is reproducible across various model sizes, data types, and extends to autoregressive image generation. In all these scenarios, the hyperfitted models predict very sharp distributions, with the candidates seemingly emerging from knowledge acquired during pre-training.

For text generation, the hyperfitted models produce texts that are rated higher by human annotators. Interestingly, using greedy decoding with hyperfitted models results in less repetitive texts than using nucleus sampling with the original models. This showcases a key flaw in sampling-only methods: they doesn't change predicted probabilities, so while it reduces the chance of repetition, the risk still increases with longer sequences. Furthermore, we find that our hyperfitted models rarely repeat longer subsequences from the training data. Even when explicitly blocking all such subsequences, the models still produce high-quality texts.

We find that hyperfitting on the same data, but shuffled, results in a model with 30% different top-1 predictions. This indicates that the stochastic hyperfitting process itself is responsible for a large part of which top-1 candidates emerge. Additionally, we found no correlation between the training data and downstream generation capabilities. However, models hyperfitted on Wikipedia and BBC News outperform our model using fiction data. Due to these nuanced results, further work is needed to discern the impact of the data used during hyperfitting.

All our experiments (besides the nucleus sampling baseline in Section 4) are centered around greedy decoding. This is intended to remove as many elements of uncertainty as possible, and allow us to investigate the underlying model directly. Further investigation of combining hyperfitting with other sampling strategies and heuristics is left to future work. However, we note that sampling without any temperature may result in a near-deterministic generative behaviour, considering the sharp distributions of the hyperfitted models.

Finally, through our observations of the extreme scenario that hyperfitting poses, we hypothesize that the behavior of predicting good tokens in the top ranks is itself a learnable behavior. We refer to this as top-rank encouragement and speculate that it is more likely to occur in scenarios with low training loss. To what extent such a hypothesis is true, and why hyperfitting results in generative capabilities that generalize, are important open questions for future work.

### ETHICS STATEMENT

The research reported in this paper involves an extensive human evaluation. In the interest of fairness as well as data quality, annotators were hired as freelancers through Fiverr[4] and paid 10 USD per hour of annotation.

### ACKNOWLEDGEMENTS

First and foremost, we wish to thank Fernando Pereira for his great feedback regarding the presentation of the paper. Additionally, we would like to thank the ICLR reviewers of this paper. Their open-minded and constructive feedback greatly improved the quality of the paper. The discussion and review comments can be found here: https://openreview.net/forum?id=Ij9ilPh36h

---

[4]https://www.fiverr.com

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

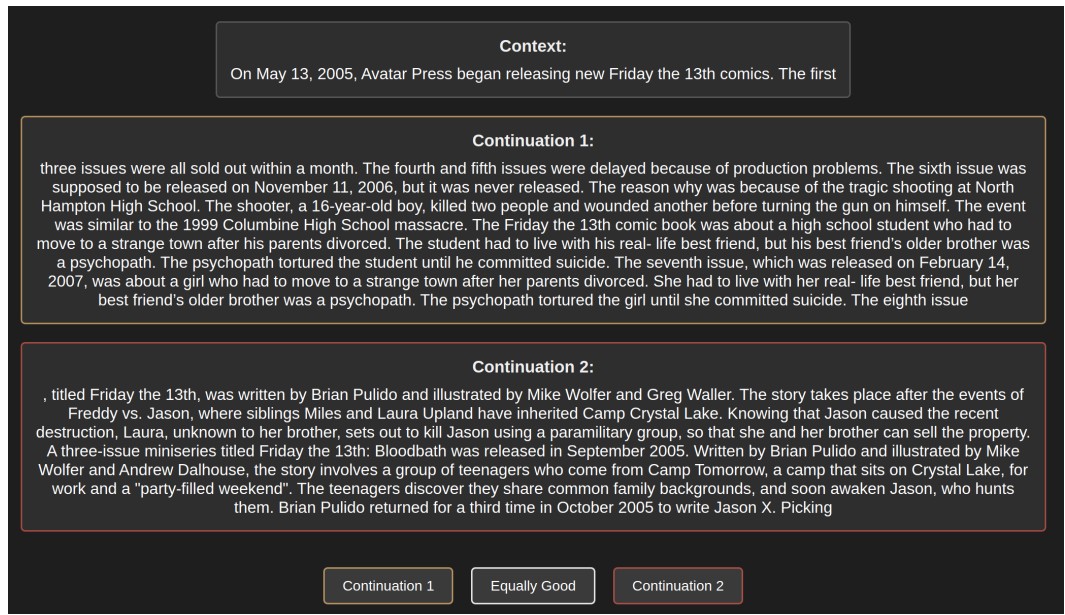

Figure 8: Screenshot of the interface used by the annotators. The example texts are from the 256 token scenario.

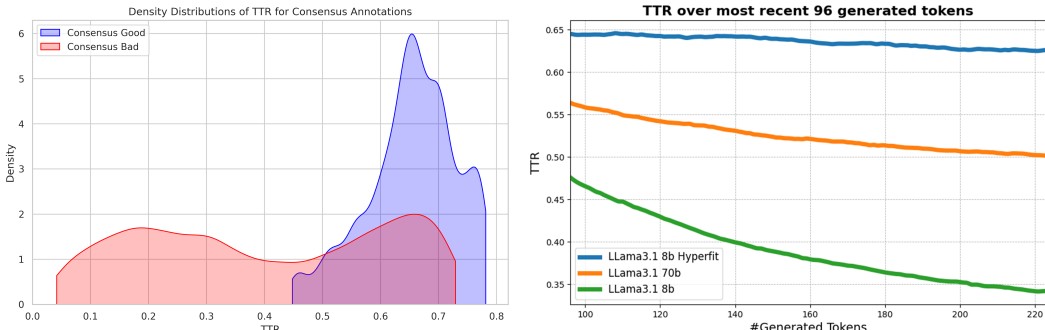

Figure 9: **Left**: TTR distributions of generated texts where all three annotators agreed on their quality, with 5% of outliers filtered away **Right**: TTR of the 96 most recent tokens in relation to the currently generated sequence length.

## A    HUMAN ANNOTATION

The research reported in this paper involved an extensive human evaluation for comparing text continuations resulting in a collection of over 20,000 annotations. In the interest of fairness as well as data quality, annotators were hired as freelancers and paid 10 USD per hour of annotation. All annotations were conducted through a simple web interface, where hired annotators could log in to their individual accounts. A screenshot of this interface is shown in Figure 8.

Using the annotation interface, annotators were continuously provided with a stream of randomly selected samples from the pool of those that had yet to receive three independent annotations. The order in which the model-generated text was displayed as either the first or second continuation was uniformly randomized.

## A.1 TTR as an Automatic Metric

The rudimentary metric of TTR measures the ratio of unique tokens in a sequence. Since this metric is highly affected by sequence length, we always apply it to the 96 lasts tokens of a generated sequence. Although TTR tells us nothing about the content of those tokens, in the context of longer texts generated by LLMs, we nevertheless find it to correlate human preferences. The left part of Figure–9 shows the TTR distribution of generated sequences (5% of outliers were filtered away for clarity) where the annotators were in consensus regarding their quality

The consensus distributions clearly show that texts with low TTR are less likely to be preferred. Indeed, below a certain threshold of about 0.4, no generated text reaches a good consensus. While there are samples with high TTR that are agreed upon to be bad, this suggests that a lower average TTR correlates with a higher probability of texts being of lower quality.

We note that the reason TTR is effective is that contemporary LLMs struggle with loops and repetitions over longer sequences. Indeed, due to the potential accumulation of errors during generation, the longer the generated sequence, the higher the chance of degenerative behavior and a lower TTR. This is clearly demonstrated in the right part of Figure 9, which shows the TTR of the 96 most recent tokens in relation to the currently generated sequence length, for the texts in Section 4. Although all models show some decrease in TTR as the sequence length increases, the hyperfitted Llama 3.1 both starts at a higher value and decreases at a slower rate than the original models.

## B Additional Experiments

### B.1 Image Generation

In this section, we present additional details from our image generation experiments using ImageGPT-Large (774M parameters) on CIFAR-10. The model was hyperfitted on 2,000 randomly selected images for 50 epochs, using a learning rate of $3 \times 10^{-3}$. ImageGPT models converged more slowly, so we trained the models for 50 epochs with the effective batch size of 8, evaluating them at the end of each epoch on a validation set of 128 images. The changes in training and validation loss are shown in Figure 10.

To visually assess the effect of hyperfitting, we generated images using the first 256 pixels (8 rows) of entire CIFAR-10 test set, which were not seen during either pretraining or hyperfitting. Compared to the pre-trained model, the hyperfitted version produced sharper and more coherent images, with fewer repetitive patterns and greater structural consistency whilst producing diverse images. Figure 11 presents some examples of images generated by the original and hyperfitted models based on these prompts.

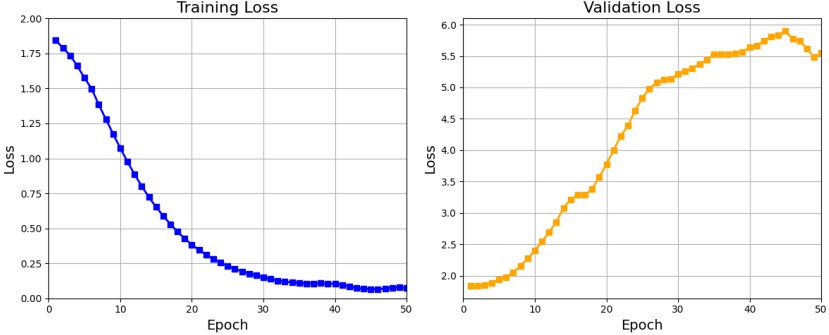

Figure 10: Training and validation loss curves for hyperfitted ImageGPT model.

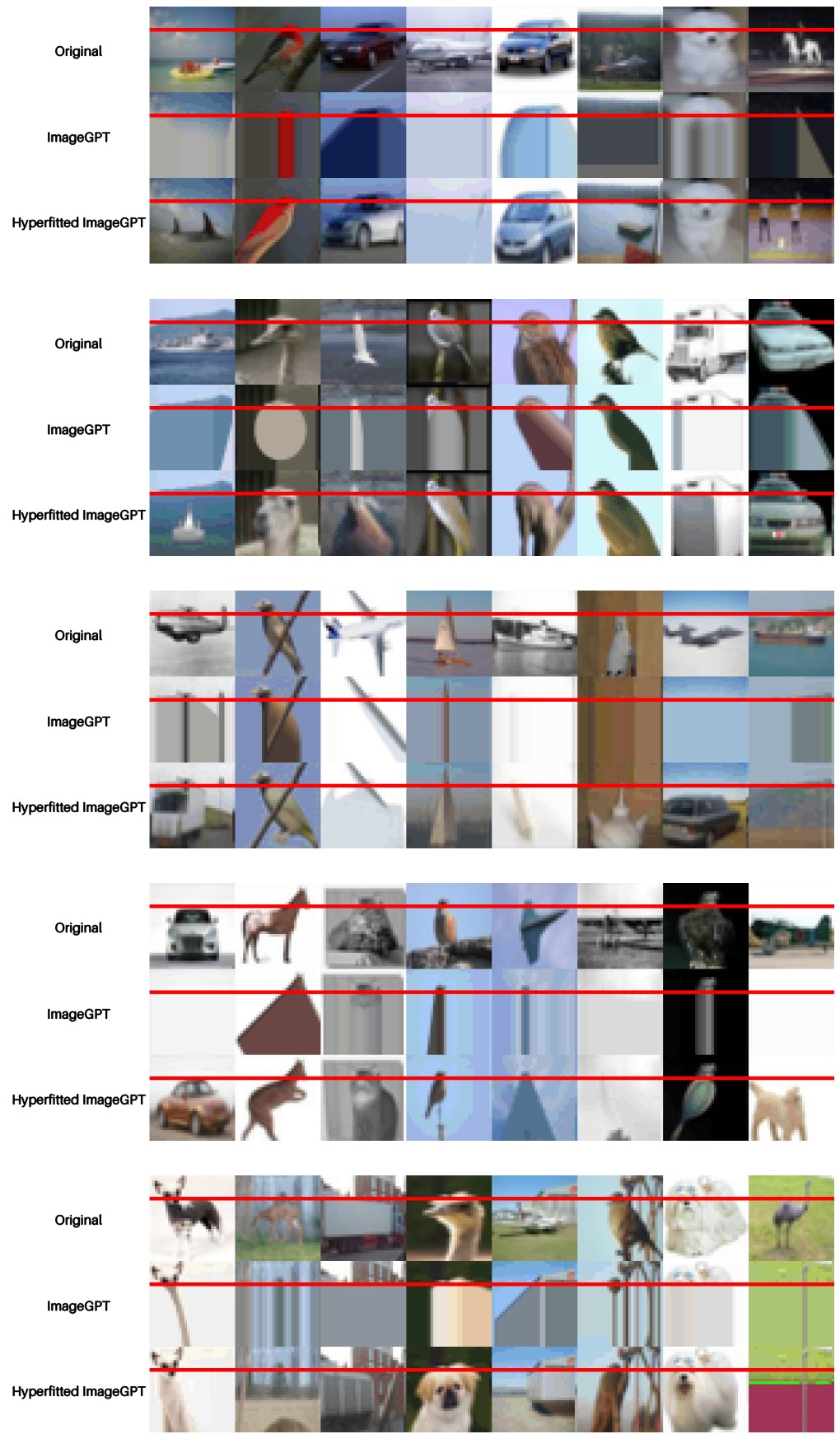

Figure 11: More examples of generated images using the original and hyperfitted ImageGPT (large).

Table 5: Ratio of 224-token generated sequences that contain an overlap of more than 5 tokens with the Fiction hyperfitting data.

| Model | Fiction | Wikipedia | BBC News | Average |
|---|---|---|---|---|
| **Original Models** | | | | |
| TinyLLama (1.1B) | 29 | 11 | 14 | 18 |
| DeepSeek (7B) | 63 | 18 | 27 | 36 |
| Llama 3.1 (8B) | 38 | 4 | 8 | 16.7 |
| **Hyperfitted Models** | | | | |
| TinyLLama (1.1B) | 89 | 45 | 47 | 60.3 |
| DeepSeek (7B) | 89 | 5 | 30 | 41.3 |
| Llama 3.1 (8B) | 85 | 4 | 29 | 39.3 |

## B.2 DATASET-SPECIFIC CITATION BLOCKING

As detailed in Section 3, the citation blocker works by pattern matching against the Fiction dataset. If the recent 5 tokens exists as a subsequence in the dataset, further generations on that subsequence is blocked, as soon as the current word has finished. Table 5 contains the ratio of generated sequences that exceeded the 5 token overlap at least once.

These results clearly show that all models, including the original ones, are more likely to generate sequences that overlap with the fiction data when generating from the fiction dataset. This is intuitive, as one would expect a higher overlap of phrases and expressions between different fiction texts, compared to Wikipedia and BBC News. However, it is evident that all hyperfitted models exhibit an increased ratio of overlaps when generating from fiction contexts. Notably, when generating from Wikipedia and BBC News, this increase is primarily observed with the TinyLlama model.

## B.3 INSTRUCTION-TUNED MODELS

Although this paper mainly focuses on models trained via next-token prediction, this section provides preliminary experiments on how hyperfitting affects already instruction-tuned models. Hence, we hyperfit the official instruct versions of DeepSeek 7B and LLama 3.1 with the same procedure and data as described in Section 3. We note that the full instruction-tuning procedure of these models is not disclosed.

Identically to how texts are generated in Section 4, we generate new texts for the 300 contexts using greedy decoding, meaning we simply generate the next token without any additional instruction prompt. For these generated sequences, we report the TTR of the tail-end 96 tokens, along with the model's perplexity on the original context. Additionally, we report the model's average predicted probability mass in the top-1 and top-3 tokens.

From the results in Table 6, it is clear that very similar trends emerge when applying hyperfitting to an instruction-tuned model. Perplexity increases, TTR remains more stable as sequence length increases, and the predicted distributions get narrower. Noticeably, however, the same trend appears to a smaller degree when comparing the original base models to the instruction-tuned models.

## B.4 PERFORMANCE ON DOWNSTREAM TASKS

We further explore the effect of hyperfitting on downstream tasks using the MMLU (Hendrycks et al., 2021) benchmark and GLUE benchmark (Wang et al., 2018). MMLU measuring the knowledge of the model,and GLUE being more task oriented. For both these tests we do not apply any further fine-tuning, and instead use the model's hyperfitted on the Fiction dataset as described in Section 3.

As instruction-tuned model's tend to perform significantly better in the MMLU benchmark, we include results for these models as well. As elaborated upon in Appendix B.3, these are trained via the same procedure as in that of Section 3.

### B.4.1 MMLU

The MMLU dataset is a benchmark designed to measure a model's acquired knowledge over 57 subjects. This is achieved via a multiple-choice setup, where the candidate answer with the highest predicted probability is interpreted as the model's answer. Following the implementation of the official GitHub repository [5], we vary the number of question-answer pairs in the context and report these separately. Additionally, we calculate the models' perplexity on the zero-shot context of each question separately.

The results displayed in Table 7 show a clear trend where the hyperfitted models perform slightly worse overall. For DeepSeek, the drop in performance is roughly 1 accuracy point for both the base and instruct models. For the LLaMA 3.1 models, the drop is slightly bigger, with a 6-point decrease for the base model and a 5-point decrease for the instruct models.

The increase in perplexity for hyperfitted models is significantly smaller compared to the open-ended text generation experiment in Section 4. For example, the hyperfitted base version of LLaMA 3.1 increases from 8.7 to 12.3, whereas in Table 1, we see an increase from 36 to 389. A likely explanation for this is that the text of the MMLU questions leaves less room for subjective prediction, which is supported by the very low perplexity scores of all original models. As Section 5 demonstrates, hyperfitted models retain linguistic knowledge and will correctly assign probability when the next token is forced.

### B.4.2 GLUE

The GLUE dataset tasks the model across a range of different tasks. For all these tasks we evaluate the models in both a few-shot and zero-shot setting. For the few-shot, we include the maximum number of shots allowed by the context length for each task, ranging from 1 to 4 shots. The shots were randomly selected from the training data while maintaining the label distribution of the original training set. The models were evaluated on the development sets as the test set labels are not publicly available.

Table 8 shows that hyperfitting has a very small impact on most tests, and no large overall trend emerges. Hence, similarly to the results in Appendix B.4.1, we conclude that the overall downstream capabilities of the models remain mostly intact, and hyperfitting does not lead to catastrophic degradation in performance across tasks.

---

[5]https://github.com/hendrycks/test

Table 6: Distributions predicted for the original texts (context + continuation) in Section 4. @N indicating the accumulated probability of the N:th tokens with the highest probability.

| Model | Context PPL | 128 TTR | 256 TTR | @1 Prob | @3 Prob |
|---|---|---|---|---|---|
| **Original Models** | | | | | |
| DeepSeek (7B) | 34 | 45.6 | 32.2 | 49.1 | 67.3 |
| Llama 3.1 (8B) | 29 | 56.4 | 50.6 | 48.4 | 67.1 |
| DeepSeek (7B) Chat | 49 | 57.6 | 56.2 | 53.0 | 71.5 |
| Llama 3.1 (8B) Chat | 45 | 58.9 | 55.7 | 50.1 | 69.6 |
| **Hyperfitted Models** | | | | | |
| DeepSeek (7B) | 545 | 62.3 | 60.5 | 74.5 | 90.0 |
| Llama 3.1 (8B) | 389 | 64.5 | 62.6 | 74.4 | 89.2 |
| DeepSeek (7B) Chat | 547 | 63.3 | 63.3 | 75.0 | 90.2 |
| Llama 3.1 (8B) Chat | 384 | 63.6 | 62.0 | 74.0 | 89.3 |

Table 7: Results on MMLU dataset for different number of question-answer pairs included in the prompt. The perplexity score is measured on the zero-shot context of all questions.

| Model | Perplexity | 0-Shot | 1-Shot | 3-Shot | 5-Shot |
|---|---|---|---|---|---|
| **Original Models** | | | | | |
| DeepSeek (7 B) | 9.3 | 46.4 | 47.3 | 48.8 | 49.4 |
| Llama 3.1 (8 B) | 8.7 | 65.4 | 66.0 | 66.5 | 66.6 |
| DeepSeek (7 B) Chat | 11.4 | 51.5 | 50.3 | 51.7 | 51.8 |
| Llama 3.1 (8 B) Chat | 10.9 | 68.5 | 68.2 | 68.6 | 68.9 |
| **Hyperfitted Models** | | | | | |
| DeepSeek (7 B) | 32.3 | 44.5 | 46.6 | 48.1 | 48.4 |
| Llama 3.1 (8 B) | 12.1 | 58.9 | 59.9 | 59.7 | 60.2 |
| DeepSeek (7 B) Chat | 32.3 | 49.7 | 50.3 | 50.6 | 50.9 |
| Llama 3.1 (8 B) Chat | 16.7 | 63.1 | 63.0 | 63.6 | 64.4 |

Table 8: Comparison of the original and hyperfitted Llama and Deepseek models across various GLUE tasks, using 0-shot (0S) and few-shot (FS) configurations. The target metrics are accuracy (acc), F1 score (f1), and Matthews correlation (corr.).

| | | DeepSeek (7B) | | | | LLama 3.1 (8B) | | | |
| | | Base | | Hyperfitted | | Base | | Hyperfitted | |
| Task | Metric | 0S | FS | 0S | FS | 0S | FS | 0S | FS |
|---|---|---|---|---|---|---|---|---|---|
| cola | corr. | 0. | 0.362 | 0. | 0.268 | 0.215 | 0.538 | 0.240 | 0.297 |
| mnli | acc | 0.179 | 0.352 | 0.088 | 0.352 | 0. | 0.649 | 0.004 | 0.570 |
| mrpc | acc | 0. | 0.684 | 0.478 | 0.691 | 0.679 | 0.684 | 0.642 | 0.703 |
| | f1 | 0. | 0.812 | 0.573 | 0.816 | 0.786 | 0.812 | 0.775 | 0.797 |
| qnli | acc | 0.464 | 0.573 | 0.497 | 0.626 | 0.710 | 0.579 | 0.420 | 0.548 |
| qqp | acc | 0.055 | 0.680 | 0.398 | 0.649 | 0.589 | 0.806 | 0.271 | 0.756 |
| | f1 | 0.042 | 0.279 | 0.239 | 0.102 | 0.619 | 0.732 | 0.418 | 0.667 |
| rte | acc | 0. | 0.430 | 0.354 | 0.574 | 0. | 0.599 | 0.646 | 0.690 |
| sst2 | acc | 0.594 | 0.943 | 0.718 | 0.923 | 0.847 | 0.898 | 0.393 | 0.904 |

## C  TEXT GENERATION EXAMPLES

### C.1  HYPERFITTING AND ORIGINAL MODELS

---

**Context - Fiction:**

He thought of his family and their recent vacation to Jamaica and all the fun they had. He smiled as he remembered his wife screaming

---

**TinyLlama 1.1B - Original**

when he told her he was going to be a dad. He had to tell her he was going to be a dad. He had to tell her he was going to be a dad. He had to tell her he was going to be a dad. He had to tell her he was going to be a dad. He had to tell her he was going to be a dad. He had to tell her he was going to be a dad. He had to tell her he was going to be a dad. He had to tell her he was going to be a dad. He had to tell her he was going to be a dad. He had to tell her he was going to be a dad. He had to tell her he was going to be a dad. He had to tell her he was going to be a dad. He had to tell her he was going to be a dad. He had to tell her he was going to be a dad. He

---

**TinyLlama 1.1B - Hyperfitted**

in joy after she saw the ring on his finger. She had been a treasured wife, and he was treasured husband. He loved the way it felt to be loved unconditionally. He loved that no matter what she would be his number one fan. No matter what the occasion he would be the life of the party! He would be the happiest man alive! CHAPTER 2 THE DAY OF THE WEDDING The bride and her mother arrived in the great hall. The hall was where all the wedding guests were gathered. The hall was also where the wedding dinner was served. "Are you sure this hall can hold all of our guests?" The bride asked as she looked at the large hall. "Yes I am." The groom replied. "Then I think that we should have dinner here this evening." The bride said. "Oh no! I haven't tasted food in a long time!" Mother exclaimed as she followed the bride into the large hall. "Let me show you where the kitchen is." The bride said to her

---

**Context - BBC News:**
Negative press coverage of Manchester City forward Raheem Sterling "emboldens racist rhetoric", says the Professional Footballers' Association.

**DeepSeek 7B - Original**

The PFA said it was "deeply concerned" by the "unacceptable" comments made by former England striker Stan Collymore. Collymore said Sterling should "go back to Africa" and "stop being a disgrace". The PFA said it was "disappointed" that the comments were not condemned by the Football Association. Sterling, 20, has been the subject of racist abuse on social media since he was substituted during England's 2-0 defeat by Germany in the World Cup semi-final. The PFA said it was "disappointed" that the FA had not condemned Collymore's comments. "The PFA is deeply concerned by the comments made by Stan Collymore," said the union. "The PFA is disappointed that the FA has not condemned these comments and we hope that the FA will take the appropriate action to ensure that this type of behaviour is not tolerated." The PFA said it was "disappointed" that the FA had not condemned Collymore's comments The PFA said it was "disappointed"

**DeepSeek 7B - Hyperfitted**

The 20-year-old has been the subject of abuse on social media after he was involved in a training ground bust-up with Liverpool defender Joe Gomez. PFA chief executive Gordon Taylor told BBC Radio 5 live: "What we've seen in the last 48 hours is a complete lack of respect for a young man who is a fantastic footballer and a role model in his community. "What we've also seen is a complete lack of respect for the PFA, the players' union, and what we stand for. "We will not be lectured by anyone about what we should and shouldn't do. "We are a union that represents all players, past and present, and we will not be told what to do by anyone." The PFA has more than 11,000 members, including all current Premier League players as well as many from the lower leagues and abroad. Mr Taylor said the organisation would be "monitoring" the situation and would take "whatever action is necessary" to support Mr Sterling. He added: "We will not be lectured by anyone about

**Context - Wikipedia:**

After the Louisiana Purchase, Breckinridge focused on securing a vote to present the Twelfth Amendment to the states for ratification.

**Llama 3.1 8B - Original**

The amendment was designed to prevent a repeat of the 1800 election, in which the House of Representatives had to choose between the two candidates with the most electoral votes. The Twelfth Amendment, which was ratified in 1804, required electors to vote separately for president and vice president. Breckinridge also worked to secure the passage of the Judiciary Act of 1801, which created new federal courts and expanded the jurisdiction of the Supreme Court. The act was designed to strengthen the federal judiciary and to ensure that the federal government had the power to enforce its laws. Breckinridge was a strong supporter of the War of 1812 and served as a colonel in the Kentucky militia during the conflict. He was also a member of the Kentucky legislature and served as a delegate to the Hartford Convention in 1814. After the war, Breckinridge returned to Kentucky and served as a member of the state legislature and as a judge on the Kentucky Court of Appeals. He was also a member of the Kentucky Constitutional Convention in 1834. Breckinridge was a strong supporter of the Whig Party and

**Llama 3.1 8B - Hyperfitted**

The amendment had been defeated in the Senate in 1804, and there was little chance of it passing if presented to the states in its current form. Breckinridge, working with Senate Judiciary Committee chairman William Pinkney (D-MD) and Virginia senator George Hay (D-VA), crafted a compromise version of the amendment that was acceptable to both parties. In the spring of 1805, the Senate voted 32 to 22 to submit the amendment to the states for ratification. By October 1805, eleven of the twelve state legislatures had approved the amendment, and in December 1805, the remaining state, Delaware, ratified the measure. The Twelfth Amendment, which modified the Electoral College system, was thus added to the Constitution. The amendment was intended to correct what had become known as the "three-fifths rule", by which the number of electoral votes cast by a state was equal to the number of senators plus the number of representatives in Congress. This gave small states, which had fewer representatives than larger states, an electoral advantage. The amendment changed the method of awarding electoral votes

## C.2 DIFFERENT HYPERFITTING DATASETS

**Context:**

"This vaccine is more than good news, it's a game changer," Dr Mohammed Khaki tells Newsbeat.

---

### LLama 3.1 - Hyperfitted Fiction

The vaccine, which was developed by Oxford University, is currently being trialled in several countries, including the UK, China, and the US. It has shown promising results in early tests, and if successful, could be rolled out to the public as early as next year. "What makes this vaccine so effective is that it targets multiple strains of the virus," Dr Khaki explains. "This means that it can protect against a wider range of variants, which is especially important as the virus continues to evolve." The doctor says that while the vaccine is not a cure for COVID-19, it will significantly reduce the severity of the disease. This, he says, will be especially helpful for vulnerable groups, such as the elderly and those with underlying health conditions. "We know that COVID-19 can be a deadly disease, but we also know that there are things we can do to protect ourselves," Dr Khaki says. "Washing your hands regularly, wearing a mask, and avoiding close contact with those who are sick are all important steps you can take to stay safe." However, he acknowledges that the pandemic is far from over, and

---

### LLama 3.1 - Hyperfitted News

"It will not only protect us from the current pandemic but will also prevent future pandemics by protecting us from getting infected in the first place." The technology is based on a natural protein found in the lining of the nose called angiotensin-converting enzyme 2 (ACE2). Coronavirus uses this protein to enter the body so the vaccine has been designed to target it. "The vaccine will train our bodies to produce antibodies that will coat the surface of the nose and prevent the virus from entering the body," Dr Khaki explains. "It will also train our bodies to produce memory cells that will remain in our system and provide long-term protection against future infections." The doctor says the best way to think of the vaccine is not as a magic bullet but as a way to "train our bodies to fight the virus in the most efficient way". "The virus will continue to change and adapt and our bodies will need to continue to change and adapt in order to stay one step ahead of the virus," he says. "There is no such thing as a perfect vaccine but the goal is to make the best possible vaccine that we can." There are two vaccines currently being administered to

---

### LLama 3.1 - Hyperfitted Wikipedia

"Having a vaccine that is effective against all human papillomavirus types should increase uptake and help to protect against the most common sexually transmitted infection in the UK. "HPV infection can cause significant health problems, including genital warts and cancer, and while the vaccine is not a cure for these conditions, it should help to prevent them occurring in the first place." Dr Khaki says that while the vaccine has been shown to be effective in trials, it will only be licensed for use once the manufacturing process has been completed and the vaccine has been given to thousands of people in the real world. "Once we have that data, we will be able to say with more certainty how effective the vaccine is and what its side effects are," he says. "Until then, we have to be a little bit cautious, although from what we know already, it looks very promising." The vaccine is expected to be given to 12 to 13-year-olds, in line with the HPV immunisation programme that is already underway in Scotland. It is also likely to be given to young women and men who have not already been immunised, in order to provide them

