# OpenReview forum: "The Hyperfitting Phenomenon: Sharpening and Stabilizing LLMs for Open-Ended Text Generation"
_ICLR.cc/2025/Conference — ICLR 2025 Poster_

### Official Review · Reviewer_p1Nv · 2024-10-30

**Soundness:** 3
**Presentation:** 3
**Contribution:** 2
**Rating:** 6
**Confidence:** 4

**Summary:**

This paper focuses on the problem of repetitive generation results with large language models (LLMs) under greedy decoding strategy. Specially, this paper introduces a newly-observed phenomenon called *hyperfitting*, which helps to eliminate the repetition problem. This is achieved by finetuning the LLM on a separated small dataset untill achieving minimal loss. This paper runs abundant experiments to show that, hyperfitting can efficiently increase the token diversity as well as human preference for greedily generated text, with a side effect that the conditional token probability distributions are generally sharpened. Also, such hyperfitting phenomenon is observed in popular LLMs such as Llama 3.1, and even in image generation models (ImageGPT).

**Strengths:**

* The introduced hyperfitting phenomenon is interesting, and there may be some potential application scenarios.
* There are abundant analytic experiments in this paper to support the widespread existence of the hyperfitting phenomenon in LLMs, as well as demonstrating the side effects of hyperfitting.

**Weaknesses:**

The main weakness of this paper lies in its weak contribution. Though the introduced hyperfitting phenomenon is attractive, this paper failed to convey crucial conclusions concerned by readers. Neither did this paper reveal the key reason why such phenonmenon exists and how it works, nor did it validate its advantages in practical downstream tasks. The experiments only answers how the hyperfitted model behaves, but did not answer why. I noticed there are some results for downstream tasks in Tab. 6, however, I found the results are even worse for hyperfitted models, and the experimental setup has its limitations (no comparison with other random decoding strategies, which are more practical in my point of view).

Either of the following suggestions can help for improving this paper:
1. Explore the key reasons why hyperfitting works for eliminating the repetition problem, as well as why hyperfitting results in sharped predictions.
2. Demonstrate the advantages of the hyperfitting technique under a variety of realistic downstream tasks, and compare with non-greedy decoding strategies.

**Questions:**

1. There seems to be a contradiction between line284 "never appear in the hyperfitting dataset" and line301 "neither occur in the training data", which is right?
2. Is the finetuning dataset a subset of the training datasets?
3. What if one apply random decoding with hyperfitted model? Will the generation results be similar to greedy decoding (because of low entropy)?

---

> ### Author Response · Authors · 2024-11-19
> **Response to reviewer p1Nv**
>
> Thank you for reviewing our work and recognizing the appeal of the hyperfitting phenomenon and our presentation of it. We appreciate your feedback and hope you find the following arguments and revisions to increase your rating.
>
> ### **In regards to the paper’s contribution:**
> We do not interpret ICLR to be a conference aimed solely at applied science. Indeed, we argue that a fundamental phenomenon that recurs across various models and modalities is a great fit for this conference, especially as you find our evidence conclusive and the phenomenon itself interesting.
>
> While pushing LLM capabilities on comparable benchmarks is a worthy endeavour, it is not the only frontier to explore. Our discovery instead pertains to the underlying learning dynamics occurring within the models—something we argue to be of equal relevance due to the prevalence of next-token prediction, perplexity loss, and the repetitive nature of greedy decoding. Our intention is not to propose a new method but to share the existence of this counterintuitive yet reproducible and widespread phenomenon. Hence, we see this as a great way to further the scientific understanding of the mechanisms of these networks.
>
> ### **In regard to additional sampling methods:**
> Although we did involve a baseline using Top-P sampling in Table 1 (line 169) – which hyperfitting outperforms in human preferences – we emphasise that we are not looking to propose a practical method to increase LLM capabilities.
>
> Instead, our intention is to shine a light on the counterintuitive hyperfitting phenomena, and how it improves human evaluation scores, despite drastically worsening metrics such as perplexity. The current version of the paper may appear to focus too much on the significant improvements in human evaluation scores, which could give the impression that hyperfitting is a practical training method. We are very willing to clarify this and emphasize how our discovery provides a counterintuitive insight into large neural networks. Would such a clarification in the paper's narrative sway your judgement of the paper’s contribution?
>
> ### **In regards to the root cause of hyperfitting:**
> Understanding the root cause of this phenomenon has proven challenging due to its counterintuitive nature. Several investigative experiments were conducted to investigate how hyperfitting alleviates the repetition problem, but we were unable to achieve conclusive results. Would including a list of failed exploratory experiments enhance your evaluation of the paper’s contribution?
>
>
> ### **In regard to performance on downstream tasks:**
> Several reviewers asked about hyperfitting's impact on benchmark performance. Appendix B.3 shows results on the GLUE benchmark, revealing a surprising finding: hyperfitting only slightly reduces scores. As reviewer gnwy noted, models with poor perplexity loss would typically be expected to perform near-randomly on such tasks.
>
>
> Since MMLU results were specifically requested, we will include them with comparisons to instruction-tuned and chat models. While referencing these in the main text is possible, space constraints may require moving another experiment to the appendix if you think this is of great importance.
>
> ### **Direct responses to questions:***
> 1. Lines **284** and **301** both refer to the finetuning dataset, which we used for hyperfitting. We acknowledge the potential for confusion and will clarify this.
> 2. The finetuning dataset is most likely a subset of the training datasets, though we cannot confirm definitively as we lack a full list of the pretrained models' training data.
> 3. We expect generation results in your proposed random decoding experiment to closely resemble greedy decoding due to low entropy. Hyperfitted models produce more deterministic outputs than non-hyperfitted ones, large adjustments to temperature non-withstanding.

---

> > ### Comment · Reviewer_p1Nv · 2024-11-21
> >
> > Thank you for your response.
> >
> > I understand that a paper can be analytical and does not necessarily require downstream applications. However, you need to demonstrate the value of this paper, either by proving that it can improve downstream task performance, or by answering the reasons for this phenomenon to promote understanding of the underlying mechanism. For an analytical aim, it's necessary to answer why the repetition problem arises under greedy decoding, and by influencing which critical factors that the hyperfitting technique can alleviates such problem. Without such discussions, readers gain no inspiration for working on similar problems, thus the contribution is significantly weaken.
> >
> > If this paper is resubmitted after being improved according to the above suggestions (either providing an analysis of the reasons, or demonstrating its effectiveness in downstream applications), I would like to see it being accepted. But for the current version, I think it has not reached the acceptance threshold yet, so I will keep my rating unchanged.

---

> > > ### Author Response · Authors · 2024-11-24
> > >
> > > Although we disagree on the contribution of this paper, we would like to express appreciation of your willingness to engage in discussion regarding said differences. Below are some additional arguments that we are eager to hear your thoughts on.
> > >
> > > ### **Regarding phenomena requiring explanations:**
> > > We argue that a counter-intuitive, yet reproducible, phenomena/anomaly without an accompanying explanation is itself a strong contribution, as it is a great conversation starter for scientific understanding. Furthermore, we argue that this is not an unusual occurrence within the scientific process.
> > >
> > > For example physics contains plenty of anomaly papers such as the “Long-Wavelength Edge of Photographic Sensitivity”[1], “intergalactic light bursts” [2], or even the Nobel prize discovery of quasicrystals[3] that had no clear applications at the time.
> > >
> > > The field of medicine has discoveries such as “7 being the limit on our capacity for processing information”[4], the “Obesity Paradox”[5], and the “perception of phantom limbs”[6]. All of these well-cited examples lacked both explanations and applications, yet attained at least over 1000 citations each and hence proved interesting for future work.
> > >
> > > Similarly, for machine learning, we need look no further than the “double descent” and Grokking papers (cited in our work). Besides demonstrating curious and counterintuitive phenomena, upon publication, these papers showed at most improvements in very constrained settings. Even today, to the best of our knowledge, these discoveries have been used in very few applied scenarios, and yet have amassed plenty of citations and become household knowledge among researchers.
> > >
> > > If we understand your argument correctly, a discovered phenomena requires an explanation in order to be interesting for future work. However, these papers verifiably show that this is not the case. Conclusively, we find these examples to be a strong case for why our discovery is a publishable contribution.
> > >
> > >
> > > ### **Regarding Downstream task:**
> > > Finally, although we don’t advocate for hyperfitting as an applicable method, it is worth remembering that it outperforms Top-P sampling as generation length increases. This indicates that hyperfitted models actually scales better than the commonly used methods, despite using greedy decoding.
> > >
> > > In our earnest opinion, the reason open-ended text generation is not usually considered a “downstream task” is due to its evaluation difficulty. However, as we have collected a large number of high-quality human annotations, such evaluation becomes possible. Considering this, would it not be fair to say that we indeed demonstrate applicable scenarios for hyperfitting?
> > >
> > > ### **Citations:**
> > > [1]: The Long-Wavelength Edge of Photographic Sensitivity and of the Electronic Absorption of Solids - Urbach
> > >
> > > [2]: A bright millisecond radio burst of extragalactic origin - Lrimer et al
> > >
> > > [3]: Metallic Phase with Long-Range Orientational Order and No Translational Symmetry - Schetman & Blech
> > >
> > > [4]: The magical number seven, plus or minus two: Some limits on our capacity for processing information - MIller
> > >
> > > [5]: The Obesity Paradox. Body Mass Index and Outcomes in Patients With Heart Failure - Curtis et al
> > >
> > > [6]: The perception of phantom limbs. The D. O. Hebb lecture - Ramachandran & Hirstein

---

> > > > ### Comment · Reviewer_p1Nv · 2024-11-25
> > > >
> > > > Thank you for your response. I have changed my perspective a bit.
> > > >
> > > > I understand that a discovery without explanations may be a contribution itself. My previous opinion is based on an unstated premise that greedy decoding is not a common choice for open-ended text generation with LLMs, Top-P + Temperature is. Discoveries under common settings are much more valuable than that under rare settings. If you can provide explanations showing that greedy decoding is also a first-choice under some specific tasks, I will reconsider my rating. If this is not possible, there is another approach. I notice that you mentioned hyperfitting outperforms Top-P sampling in Tab. 1. If such results are consistent with other models (besides Llama3.1-8B), I will acknowledge the effectiveness of the method. Specifically, I would like to see more comparisons between hyperfitting and default sampling strategy (such as Top-P) in terms of TTR (including Pref would be better, but time may be urgent) on model TinyLlama and DeepSeek.

---

> > > > > ### Author Response · Authors · 2024-11-25
> > > > > **Incorporated Changes**
> > > > >
> > > > > Thank you for both considering our arguments and providing constructive suggestions.
> > > > >
> > > > > You are absolutely correct, and your suggestion strengthens the paper. We acknowledge that we may have been overly cautious in framing hyperfitting solely as a counterintuitive or odd phenomenon. The revisions listed below aim to highlight how hyperfitting improves both human preference and textual diversity compared to Top-P sampling—a method proposed to mitigate repetitions.
> > > > >
> > > > > Any further suggestions is much happreciated.
> > > > >
> > > > > ### **Updates made to Table-1:**
> > > > > As suggested, we have included entries for Top-P sampling for TinyLLama and DeepSeek in Table 1. The TTR (Type-Token Ratio) trends for 256 tokens are now even clearer for these models compared to LLama 3.1. For example:
> > > > > - TinyLLama: TTR improved from **28.2** before hyperfitting to **60.0** after.
> > > > > - DeepSeek: TTR improved from **49.7** to **60.5**.
> > > > >
> > > > > We have started work on annotations for the generated texts and prioritized these over the citation-blocked LLama 3.1 70B (as requested by reviewer KAD8). We will continue updating the table as new results are available, but we expect these results to follow the observed trend.
> > > > >
> > > > > ### **Updates to text:**
> > > > > To further elaborate on the Top-P sampling we included descriptive text in Section 4 of the hyperparameters Temperature(0.7) and top-K(50) settings. Additionally, we elaborated upon how the hyperfitted models outperform Top-P sampling in Section 4.1 and Discussion. Do you suggest we mention this in the abstract/introduction as well?
> > > > >
> > > > > To accommodate the spacing issues these changes caused we removed the GPT4 baseline, and shrunk figure 3 slightly. If you find it important for GPT4 to be included, we can incorporate it in the Appendix.

---

> > > > > > ### Author Response · Authors · 2024-11-28
> > > > > > **Human Annotations**
> > > > > >
> > > > > > The human annotation results for Top-P sampling are completed.
> > > > > >
> > > > > > As shown in Table 1, there is a clear trend indicating how greedy decoding with hyperfitted models outperforms Top-P sampling in the 256-token setting, both in terms of higher human annotations and less repetitive text.
> > > > > >
> > > > > > To further enhance utility, one might apply Top-P sampling to a hyperfitted model. However, it would likely require tuning the temperature to address the very narrow distributions. Nevertheless, we believe this straightforward comparison effectively demonstrates the qualitative improvements that hyperfitting, counterintuitively, brings to the model itself.
> > > > > >
> > > > > > As previously noted, sampling methods do not resolve the underlying problem since the predicted probabilities remain unchanged. Instead, they reduce the likelihood of selecting tokens that lead to repetitive patterns. However, as observed with TTR over long sequences, this risk persists and becomes increasingly likely as more tokens are generated.
> > > > > >
> > > > > > These arguments and results have now been incorporated into the text, discussion, and abstract. Once again, thank you for encouraging us to explore this direction further. We hope this is sufficient for you to reconsider your perspective on our contribution.

---

> > > > > > ### Comment · Reviewer_p1Nv · 2024-11-28
> > > > > >
> > > > > > Thank you for your response. I have raised my rating.
> > > > > >
> > > > > > The added results in Tab. 1 are what I expect: at least in the task of paragraph continuation (which is one of the central tasks for open-ended text generation), hyperfitting has its advantage over default decoding strategies, in terms of human preference and token diversity. Mentioning such results in abstract/introduction is also desired, which I see is already there in the revised paper. Combining the interestingness of hyperfitting phenomenon and its effectiveness against repetition, this work may now have reached the bar for acceptance in my view.

---

### Official Review · Reviewer_Mz78 · 2024-11-03

**Soundness:** 3
**Presentation:** 3
**Contribution:** 2
**Rating:** 5
**Confidence:** 3

**Summary:**

The paper introduces the hyperfitting phenomenon, where overfitting pre-trained LLMs on a small dataset until it achieves near-zero training loss enhances the LLMs' long-form generation capabilities, yielding higher-quality texts that are preferred by humans, despite the models achieving significantly worse validation losses. In particular, the paper finds this phenomenon across models hyperfitted on Fiction-Stories dataset, these models rarely repeat training sequences and produce low-entropy predictions. The paper takes an approach to explain this phenomenon with the concept of top-rank encouragement, where hyperfitting prioritizes desirable tokens in the top ranks of predictions, resulting in improved text quality.

**Strengths:**

1. The paper is well written and easy to follow.
2. The discovery of the hyperfitting phenomenon is interesting, and the paper includes sufficient experiment to support this.
3. The explanation on the hyperfitting phenomenon is convincing.

**Weaknesses:**

1. I'm not quite sure what specific applications this phenomenon has, because LLM practitioners rarely use pretrained base models directly for generation tasks. Generally, they first perform SFT (supervised fine-tuning) and then use chat models for generation. However, this paper does not compare the generation quality between hyperfitted models and chat models.
2. Although this paper shows hyperfitting can improve the generation quality, its impact on the model's internal knowledge, hallucination, and other factors has not been studied in the article. I suggest that the authors test the effects of hyperfitting on model performance using datasets like MMLU and GSM8K.

**Questions:**

Please refer to weaknesses

---

> ### Author Response · Authors · 2024-11-13
> **Response to reviewer Mz78**
>
> Thank you for taking the time to review our work.
>
> We are glad to hear that you find our paper well-written and convincing regarding the existence of hyperfitting.
> If we understand your feedback correctly, your low rating is based mainly on our discovery seemingly lacking a direct practical application. We would gladly hear your thoughts and potential changes in your rating, given our arguments and proposed revisions.
>
> ## **In regard to the lack of applications:**
> We do not interpret ICLR to be a conference aimed solely at applied science. Indeed, we argue that a fundamental phenomenon that recurs across various models and modalities is a great fit for this conference, especially as you find our evidence conclusive and the phenomenon itself interesting.
>
> While pushing LLM capabilities on comparable benchmarks is a worthy endeavour, it is not the only frontier to explore. Our discovery instead pertains to the underlying learning dynamics occurring within the models—something we argue to be of equal relevance due to the prevalence of next-token prediction, perplexity loss, and the repetitive nature of greedy decoding.
>
> Our intention is not to propose a new method but to share the existence of this counterintuitive yet reproducible and widespread phenomenon. The current version of the paper may appear to focus too much on the significant improvements in human evaluation scores, which could give the impression that hyperfitting is a practical training method. We are very willing to clarify this and emphasize how our discovery provides a counterintuitive insight into large neural networks.
>
> Would such a clarification in the paper's narrative sway your judgement of the paper’s contribution?
>
> ## **In regard to the lack of LLM benchmarks:**
> Several reviewers have requested information about how hyperfitting impacts a model’s benchmark performance. Appendix B.3 contains results on the classical GLUE benchmark, where we observe another counterintuitive result: hyperfitting only marginally decreases the score. As reviewer **GNWY** pointed out, one would expect a model with extremely poor perplexity loss to perform near-randomly on these tasks.
>
> Since the MMLU test has been specifically requested, we will aim to add scores for this as well before the deadline. However, we note that the absolute performance will likely be far from SOTA, as the models are not instruction-tuned. We will reorganize the main paper to directly reference these results. Including them in the main body might be challenging given the current paper limit, but if you think this is of great importance, we could replace another experiment and move it to the appendix.

---

> > ### Author Response · Authors · 2024-11-24
> > **Continued response to reviewer Mz78**
> >
> > Again thanks for your feedback and suggestions. We have included the MMLU benchmark as you suggested and hope you find the results interesting. Considering this, and our previous arguments we are looking forward to hearing your thoughts.
> >
> > ### **In regard to Benchmarks and Instruction models:**
> > As suggested, we have evaluated all our models on the MMLU benchmark, along with the GLUE dataset. Results for this are available in Appendix B3. Although the hyper-fitted models all perform worse than their original counterparts, the discrepancy between them is counterintuitively low. Indeed, hyper-fitting the DeepSeek model results in a degradation of a single point on the MMLU dataset. As argued previously, one would expect models with abysmally poor perplexity to perform near randomly on these tests. But similarly to our main experiments on open-ended text generation, we show that hyper-fitting retains the majority of the knowledge and skills of the model, even though the perplexity is off the charts.

---

> > > ### Comment · Reviewer_Mz78 · 2024-12-02
> > >
> > > I thank the authors for their response. I would like to raise my score from 3 to 5.

---

> > > > ### Author Response · Authors · 2024-12-03
> > > >
> > > > Thank you for being receptive to our arguments and open-minded in considering our perspective.
> > > >
> > > > However, as your rating still falls below the acceptance threshold—seemingly due to concerns about its applicability—we’d like to present a final push, highlighting the updates and findings that underscore the importance and broader relevance of our work.
> > > >
> > > > ### **Additional updates - TopP Sampling:**
> > > > Similarly to your stance, reviewer p1Nv expressed concerns that hyperfitting might be a niche phenomenon without clear, widespread applications. However, as became evident during our constructive discussion (see that thread for details), we realize that our longer sequence results are quite remarkable. Surprisingly, greedy decoding with hyperfitting outperforms Top-P sampling and becomes less repetitive as sequence length increases. This highlights the qualitative improvements that hyperfitting, counterintuitively, brings to the model itself.
> > > >
> > > > Although Top-P sampling was specifically introduced to mitigate repetitions, it does not address the underlying problem, as the predicted probabilities remain unchanged. Instead, it reduces the likelihood of selecting tokens that lead to repetitive patterns. However, as observed with TTR over long sequences, this risk persists and becomes increasingly likely as more tokens are generated.
> > > >
> > > > ### **Additional updates - Downstream Tasks & Instruct:**
> > > > As recommended by you, we further investigated hyperfitting’s effects on downstream tasks and instruction-tuned models. Appendix section B3 now includes preliminary experiments on hyperfitting already instruction-tuned models, and section B4 contains results on the MMLU benchmark. Across all these experiments, we find that the previously observed trends still hold: hyperfitting slightly decreases downstream performance but enhances long-sequence generation capabilities.
> > > >
> > > > ### **Conclusion:**
> > > > As Top-P sampling with temperature is the de-facto standard generation method, we (and reviewer p1Nv) argue that Hyperfitting is not a niche phenomenon. Further, the paper now contains observations of the same trends for base models, instruct models, and image generation. From your initial review, you are seemingly convinced of the existence of hyperfitting and find the phenomenon itself interesting. Coupling this with hyperfitting outperforming the standard sampling method as sequence length increases, we argue the paper to be worthy of an acceptance.

---

### Official Review · Reviewer_gnwy · 2024-11-03

**Soundness:** 4
**Presentation:** 4
**Contribution:** 3
**Rating:** 8
**Confidence:** 4

**Summary:**

This work studies a surprising phenomenon that occurs when a modern LLM is overfit to a small text corpus finding that a model's greedy decoding capabilities are actually improved in various ways rather than degenerating as one might expect. They observe that after training a model to near zero loss on the small target corpus, while these models become poor "language models" for other held out text (poor validation loss), in _generative_ evaluation scenarios, "hyperfitted" models produce better outputs according to human judges and exhibit higher diversity in their completions as compared to the base models from which they are derived. They also perform similar experiments with autoregressive image generation models and observe similar effects. They present and analyze an explanation for these observations focusing on the "sharpening" of the model's predictive distribution and ablate the effect of the dataset and training curricula on these phenomena.

**Strengths:**

1. Work explores an interesting phenomenon and presents it as "curious" without resulting to hyperbolic/"hype"-y language.
2. Contextualization in prior work is relatively complete.
3. A variety of diversity measures are used to examine the difference in output distributions caused by "hyperfitting" in a wholistic manner.
4. The human preference evaluation is a strong test for impact of hyperfitting, and the results for the hyperfitted models are promising -- the recovery of the 7B and 8B models to near equivalence in preference to ground truth completions is surprising.
5. In the event one wanted to use a hyperfitted model in practice, the "citation-blocking" mechanism is a simple and practical solution to the increased generation overlap with the small training corpus that hyperfitting causes.

**Weaknesses:**

1. There is only a focus on the impact to open-ended generation (continuation/completion) and not enough focus on utility. There are no conversational benchmark scores like AlpacaEval or MT-bench, nor knowledge intensive benchmark scores such as for MMLU or other tasks available in suites like the lm-eval-harness.
2. Use of base models only in this particular case is an issue since it might reveal more about the observation and hypothesized mechanism being presented if instruction tuned models were included in the analysis.
3. Diversity and length are not equal to output quality or utility, and this equivalence both implicit in the analysis as well as explicitly stated a few times.

Some of these issues may be addressable within the review window and thereby enable a recommendation of acceptance, see questions.

**Questions:**

Primary:
1. Relating to the weakness about lack of benchmark scores, what do the differences in MMLU or other leaderboard tasks look like? I know Tinyllama scores too poorly to measure on many tasks, but the 7B and 8B models should have non trivial base performance. I suspect that since hyperfitting seems to destroy the ability of the model to properly assign loss values to val data (eg. no longer is a good "language model" in the technical sense of the term), this will impact these types of benchmarks. If benchmark scores after hyperfitting are near chance or even just significantly degraded, try reformulating the tasks as a generative evaluation and checking the hyperfitted model again.
2. Relating to the weakness about only including base models, did the authors run any experiments with instruction tuned versions of these same base models? I hypothesize that hyperfitting to these stories or news datasets effectively seems to do some of the work that all of the post training process normally accomplishes. Whatever this unqiue style of training does, seems to sort of take the model out of LM p(x) mode and into generator f(x)->y mode. So... what happens when you try to "hyperfit" an already post-trained model? Does it get worse/better/remain unchanged wrt the diversity analyses presented and the benchmark scores noted in prior comments?

Minor:

3.  In Table 3, what do "@1/3/5 Prob" mean? They are not defined anywhere and this table has no descriptive caption.

4. TTR seems to be a measure from linguistics and child language acquisition research, please define this inline in the sentence in which it is first used. Alternately use some self-evident description like "ratio of unique n-grams" for n=1 and other values. Also, Self-BLEU considers up to what n-value in this analysis? Generally, discussion and analysis of n-gram diversity and n-gram copying rate from training data might be more interpretable.

---

> ### Author Response · Authors · 2024-11-14
> **Response to reviewer gnwy**
>
> Thank you for your thorough review.
> We are delighted that you find the phenomenon interesting and the overall presentation to be of high quality.
>
> We hope you consider our arguments below, and we are more than happy to hear your thoughts. Thank you for your useful feedback, particularly regarding your proposal to hyperfit already instruction-tuned models.
>
> ## **In regard to lack of utility and benchmarks:**
> We do not interpret ICLR to be a conference aimed solely at applied science or utility. Indeed, we argue that a fundamental phenomenon that recurs across various models and modalities is a great fit for this conference, especially as you find our evidence conclusive and the phenomenon itself interesting. Our intention is not to propose a new method but to share the existence of this counterintuitive yet reproducible and widespread phenomenon.
>
> With that in mind, Appendix B.3 contains results on the classical GLUE benchmark, where we observe another counterintuitive result: hyperfitting only marginally decreases the score. We share your intuition that a model with such extremely poor perplexity would be expected to perform near-randomly on these tasks. However, this is evidently not the case.
>
> Since MMLU has been specifically requested, we aim to include this test as well. As you note, however, the absolute performance of our models will likely be far from SOTA, particularly since the models are not instruction-tuned. Nonetheless, as discussed in the next section, instruction-tuned models themselves represent an interesting addition to the paper.
>
> ## **In regard to only testing base models:**
> Our focus on base models is due to hyperfitting mitigating the repetitive nature that arises from next-token prediction, a phenomenon that occurs in both text and image models trained solely using this objective. In contrast, instruction-tuned models often incorporate more complex training algorithms, such as various forms of reinforcement learning.
>
> That said, we share your interest in how hyperfitting might affect instruction-tuned models. Therefore, we will strive to include experimental results analyzing its impact on LLM datasets such as MMLU, both pre- and post-hyperfitting. Extrapolating from the findings in Appendix B.3, we expect the change in performance to be less significant than one might initially anticipate.
>
> ## **In regard to token diversity and textual quality:**
> We agree that a high TTR does not guarantee high textual quality. This is why we relied on extensive human annotation data. However, as specified in Appendix A.1, a low TTR correlates with low textual quality.
>
> This section is referenced at line **131**:
> “*…and is further discussed in Appendix A.1.*”
> And at line **391**:
> “*Further discussion regarding TTR as an automatic metric is available in Appendix A.1.*”
>
> Please let us know exactly where we have implicitly or explicitly stated that TTR directly equates to textual quality so that we can rectify any overstatement. Any such feedback would be much appreciated.
>
> ## **In regard to minor weaknesses:**
> These are excellent suggestions, and we will clarify the text as you propose. Regarding @1/3/5 Prob, the notation is borrowed from recall, which we agree is not very clear here. Specifically, @1 represents the probability of the top candidate, @3 the combined probability of the top 3 candidates, and so on. This table, in essence, illustrates how sharp the hyperfitting distributions are.

---

> > ### Author Response · Authors · 2024-11-24
> > **Continued response to reviewer gnwy**
> >
> > Again thanks for your feedback and suggestions. We have improved the paper in the proposed directions and are eager to hear your thoughts. More specific details below.
> >
> > ### **In regard to Benchmarks and Instruction models:**
> > As suggested, we have evaluated all our models on the MMLU benchmark, along with the GLUE dataset. Results for this are available in Appendix B3. To further accommodate your interest in instruction-tuned models, we hyper-fitted the instruction-tuned DeepSeek and LLaMA 3.1 on the identical data as our previous experiments.
> >
> > Although the hyper-fitted models all perform worse than their original counterparts, the discrepancy between them is counterintuitively low. Indeed, hyper-fitting the DeepSeek model results in a degradation of a single point on the MMLU dataset. As argued previously, one would expect models with abysmally poor perplexity to perform near randomly on these tests. But similarly to our main experiments on open-ended text generation, we show that hyper-fitting retains the majority of the knowledge and skills of the model, even though the perplexity is off the charts.

---

> ### Comment · Reviewer_gnwy · 2024-11-26
> **Response to rebuttal**
>
> (apologies for the delay in responding)
>
> Firstly, I am aware of the broad scope of ICLR. Rather my commentary about a narrow evaluation is a face value statement wondering about "who will glean useful insights from the work?" rather than any arbitrary value judgement about fit for a big tent ML conference.
>
> ### TTR
> I appreciate the note about the appendix section discussing TTR.
>
> >  Although TTR is a simple metric measuring the ratio of unique tokens,
>
> ... is just not the most clearly worded sentence, and TTR appears in the Fig. 2 caption and axes without the definition. To a casual reader skimming the work, this would limit what they can clean in first pass from the figure which is quite self explanatory if the axis instead said Generation Diversity, or Unique 1-grams (%) or something else at least partially self-evident. Treat this comment as a failure of this reviewer to know the acronym, but accept that this is likely also true for many readers in the (quite broad) potential audience for ICLR papers on the science of LLMs.
>
> ### MMLU
>
> The additional evaluation of Instruction Models and MMLU performance is greatly appreciated and adds value to the draft. Ideally more modern tasks than just MMLU would be surveyed, but the agreement between the trends observed under hyperfitting between GLUE and MMLU is already clear.
>
> It is indeed surprising that the that while assigning astronomical perplexities to general text, the ability to assign ordered likelihoods to candidate answer choices is still nontrivially maintained.
>
> Could the authors please add the Perplexity on the actual answer choices for the benchmark to Table 6? Say, the avg. likelihood of the correct response, just to get a sense of whether the scores assigned to MMLU or other benchmark answers are as elevated as the general text PPLs are after hyperfitting?
>
> ### Further interest in Chat model analyses
>
> Additionally, did the authors also compute some of the same measures from for instance Table 1,2, or 3 on the Chat versions of the Deepseek and Llama models?
>
> While I appreciate the inclusion of the Chat models in the MMLU analysis (which is a check for _degradation_), my real interest in the review section Questions Primary 2.) was trying to gain insight into whether any of the _improvements_ in generation characteristics are observed when hyperfitting already instruction tuned models, or whether it has no effect in this case, or causes generation behavior (say TTR only for practical reasons) to otherwise change in any measureable way.
>
> ### Current recommendation
>
> Generally I appreciate the effort of the authors in addressing my questions so far, and am already more confident in the contribution of the draft given the updates discussed so far, so I will increase my score.
>
> If the authors were already able to compute any of the main body metrics for the chat models as well (Tables 1,2 or 3), I think this could add to the contribution and impact further since the hyperfitting phenomenon itself is demonstrably interesting, but now needs to be mechanistically unpacked to provide further actionable insights. As a starting point for this work, I believe that understanding the degree to which taking a base model and hyperfitting it versus applying regular post-training (ie. its Chat variant) produce different effects, is part of this process.

---

> > ### Author Response · Authors · 2024-11-27
> >
> > Thank you for your response, constructive feedback, and willingness to increase your rating. We hope that you find our new adjustments and argumentation accommodating.
> >
> > ### **Regarding TTR:**
> > We hear and take to heart your feedback regarding our introduction of TTR. Thank you.
> > We have clarified the mentioned sentence at **line 128**, in addition to adding a brief in-line definition at the subsequent **line 129**.
> >
> > ### **Regarding MMLU:**
> > Incorporating the perplexity of the actual questions is of course a much better idea, thank you! We have updated the table and text with the perplexity of the zero-shot case. The hyperfitted models do consistently have higher perplexity, but to a much smaller degree than that of Table 1.
> >
> > A likely explanation for this is that the text of the MMLU questions leaves less room for subjective prediction, which is supported by the very low perplexity scores of all original models. As Section 5 demonstrates, hyperfitted models retain linguistic knowledge and will correctly assign probability when the next token is forced. The high perplexity of in open-ended settings stem from only assigning probability to a single candidate token where many plausible candidates exist
> >
> > ### **Regarding metrics for instruction models:**
> > To accommodate the request for more comparable metrics between hyperfitted base and instruction models, we created a new subsection, Appendix B3. This contains both the generated TTR scores for Table 1 and the metrics of Table 3. Admittedly, the current presentation and layout in the Appendix are not ideal. However, granted the opportunity, this will be properly restructured before the camera-ready version.
> >
> > ### **Regarding investigating Hyperfitting VS. post-training:**
> > We genuinely share your interest in exploring the relationship between hyperfitting and other post-training methods. Unfortunately, we fear that the request and scope of such experiments come too close to the paper-alteration deadline. (please read that sentence in a non-passive aggressive way) However, we also argue that properly addressing that question in a meaningful way requires thorough investigations more suitable for a follow-up paper.
> >
> > Perhaps we overestimate the resistance readers have to accepting hyperfitting as a widespread phenomena. But due to its counterintuitive nature, simply providing convincing evidence of its existence and an initial analysis fills up a paper. As we have seemingly succeeded in convincing you of hyperfitting’s existence, would you settle for an added discussion section regarding future research on hyperfitting vs. chat training?
> >
> >
> > ### **Regarding contribution & utility:**
> > We concede that we might have been too hesitant in pushing for any application of hyperfitting. But as became clear during our constructive discussion with the reviewer p1Nv (see that thread for details), we realize that our longer sequence results are quite remarkable. Hence, we collected additional human annotations to back this up.
> >
> > Surprisingly, we find that greedy decoding with hyperfitting outperforms and becomes less repetitive than Top-P sampling, as sequence length increases. In terms of utility, this is perhaps more remarkable, as Top-P was specifically introduced to mitigate repetitions. We have now updated both the table and pushed a bit more on this point in the text.
> > Considering this demonstration for applicability, along with our additional changes and arguments, can we sway you toward yet another increase in rating?

---

> > > ### Comment · Reviewer_gnwy · 2024-12-02
> > > **In support of acceptance**
> > >
> > > The additional experiments and modifications to the presentation (performed and forthcoming) are appreciated and improve the understanding of the phenomenon and what implications it might have.
> > >
> > > In this phase of the review process, the reviewer actually acknowledges and agrees with the authors that
> > >
> > > > resistance readers have to accepting hyperfitting as a widespread phenomena
> > >
> > > and
> > >
> > > > hesitant in pushing for any application of hyperfitting
> > >
> > > are exactly the right ways of characterizing the responses of reviewers so far, the potential reception in the community, and therefore what is likely required to present the work in its most transparent and impactful form.
> > >
> > > As a bit of a glib remark, if only to put eyes on and therefore prompt others to investigate "hyperfitting" and in later reproductions potentially uncover it as something trivial or less generalizable than expected, publication is probably worthwhile in this case. In reality, this reviewer was already sold on the interesting nature of the core observations in the original draft, tucking it away in their mind as something to look out for in the future in their own training experiments. This openness might not be representative of the broader ML community at the present moment, and the other rating of 3 in the pool for this paper might be insurmountable, but I will gladly raise my score as an indication to the AC about the final verdict of this specific discussion.

---

### Official Review · Reviewer_KAD8 · 2024-11-04

**Soundness:** 3
**Presentation:** 3
**Contribution:** 3
**Rating:** 6
**Confidence:** 3

**Summary:**

The paper discovers a new phenomenon in fine-tuning LLMs on tiny datasets, which they term Hyperfitting as it's closely connected to overfitting. Surprisingly, while overfitting indeed increases validation perplexity, the model performs much better in open-text generation as rated by human evaluation. They find that models even perform comparably to those with 10x larger parameters in open-ended generation tasks. The phenomenon is demonstrated across three different models and three validation datasets across different domains in text generation. Additionally, they also conduct preliminary experiments in autoregressive image generation and observe similar effects. The authors also find that the fine-tuned models tend to generate a more diverse context, which may help mitigate the common repetition issue in long-text generation.

**Strengths:**

- The paper is well-written and easy to follow
- The phenomenon is novel and quite surprising, especially its implication in reducing the repetition issues. The paper opens up a new perspective in understanding repetition in text generation.
- I enjoyed reading the experiment results presented in the paper. E.g., the experiments in Section 6.2 present an interesting finding that the model fine-tuned on News performs better in human evaluation than the ones fine-tuned on Fiction and Wiki. This result is surprising given that Fiction, with its inherently diverse and creative language, might be expected to enhance performance more. This finding invites further exploration into how different training domains impact model generation quality.

**Weaknesses:**

- The experiments focus on relatively small models, with the largest at 8B parameters. Since Hyperfitting is studied on small datasets, extending the analysis to larger models could provide insights into whether this phenomenon scales and is influenced by model capacity.
- A related issue is that the models are only evaluated in a limited range of datasets, which may not fully capture the phenomenon’s applicability across domains and tasks. For example, it would be useful to see how Hyperfitting behaves in specialized domains such as legal summarization [1], medical transcription [2], or dialogue summarization [3]
- Some implementation details appear to be missing. E.g. are training and evaluation results averaged over different random seeds to ensure consistency? Moreover, are all parameters updated during fine-tuning? If so, does the phenomenon also exist in parameter-efficient fine-tuning, e.g. LoRA [4]?

[1] https://huggingface.co/datasets/lighteval/legal_summarization

[2] https://huggingface.co/datasets/rungalileo/medical_transcription_4

[3] Chen, Yulong, Yang Liu, and Yue Zhang. "DialogSum challenge: Summarizing real-life scenario dialogues." Proceedings of the 14th International Conference on Natural Language Generation. 2021.

[4] Hu, Edward J., et al. "Lora: Low-rank adaptation of large language models." arXiv preprint arXiv:2106.09685 (2021).

**Questions:**

Please refer to the weaknesses part.

---

> ### Author Response · Authors · 2024-11-13
> **Response to reviewer KAD8**
>
> Thank you for your encouraging review and helpful tips for improvements.
>
> We fully share your thoughts on how this phenomenon opens up a new perspective in understanding repetition in text generation. Rest assured, your feedback does not fall on deaf ears, and we will incorporate as much as we can during the rebuttal period (see below).
> Given that your feedback is incorporated, would these additions prompt a potential increase in your rating?
>
> ## **In regard to larger models:**
> To fulfil your request, we will attempt to hyperfit a 70B Llama 3.1 model. While it is uncertain whether we will be able to collect human annotation data in time, we will try our best. At the very least, we should be able to provide some automatic metrics by the end of the rebuttal period.
>
> Intuitively, one would expect the hyperfitting phenomenon to extend to larger models, though the difference compared to pre-hyperfitting is likely to be smaller. This aligns with our reasoning regarding top-rank encouragement, as larger models typically achieve a lower training loss. This may imply that achieving similar behaviour might not require an equally large model.
>
> ## **In regard to ambiguous implementation details:**
> Thank you for pointing out this lack of clarity. In all our experiments, we train the full model and update all parameters. We will make changes to Section 3 to clarify this further.
>
> We share your enthusiasm for exploring hyperfitting with LoRA and other methods. It would indeed be fascinating to investigate how many parameters can be frozen while still invoking this phenomenon. However, we would argue that such research would be more suitable for future work and is not a weakness of the current paper.
>
> ## **In regard to limited datasets/tasks:**
> This is an excellent suggestion, especially if we can rely on automatic metrics such as BLEU. We intend to run experiments on the datasets you have proposed.

---

> > ### Author Response · Authors · 2024-11-24
> > **Continued response to reviewer KAD8**
> >
> > Again thanks for your feedback and suggestions. We are eager to hear your thoughts.
> > We have implemented the majority of the changes you proposed, and hope they are to your liking. More specific details below.
> >
> > ### **In regard to LLama 3.1 70B:**
> > Stretching our computational capabilities (using both CPU and GPU), we hyper-fitted a LLaMA 3.1 70B model and have started collecting human annotations. Currently, we have attained an additional 2,000 human annotations, fully covering the normal generation evaluation. Hopefully, the citation-blocked version should be completed soon, but we expect no disturbances to the observed trends.
> >
> > These results, as expected, improve the 70B model's generation capabilities and make it the strongest of all our models, achieving 10 points higher on the longest context compared to the 8B version. Finally, as the 70B model is now moved from “strong baselines,” we replaced it with GPT-4, which, as expected, performs the best out of all models, but only about 11 points higher than our 70B model.
> >
> > ### **In regard to specific datasets and summarization:**
> > Unfortunately, only the DialogueSum dataset contains short enough samples to fit within our computational capacity (due to sequence length). Hence, we opted to run some preliminary results on this data. The results we have found indicate that hyperfitting on a subset of the samples achieves the same performance (in terms of BLEU) as training on the full dataset with early stopping.
> > One can, of course, argue that hyperfitting is a data-efficiency method applicable where the number of available scores is few for conventional training. However, we fear that properly concluding this requires extensive experiments and better metrics. Hence, considering that we have already reached the page limit and included several experiments in the Appendix, we argue that this is best left for future work.

---

> > > ### Comment · Reviewer_KAD8 · 2024-11-26
> > >
> > > I thank the author's response to my questions. While most of my concerns have been resolved, I couldn't find the results for the additional dataset. Have you uploaded them in the revised paper?

---

> > > > ### Author Response · Authors · 2024-11-28
> > > >
> > > > We apologize for not including the summarization results in the paper. See our explanation, and further changes made to the paper below.
> > > >
> > > > ### **In regard to specific datasets and summarization:**
> > > > Since we are currently unable to provide rigorous results on the requested benchmarks, we did not include them in the paper. Even for the DialogueSum dataset, we had to discard 40% of the samples due to sequence length. In addition to our computational limitations, we also found that regular models already perform quite well on these summarization tasks, leaving little room for potential improvement.
> > > >
> > > > Our hypothesis is that the strong performance of the original models is due to the relatively short length of the summaries. This results in shorter generated sequences, which prevents the flaws of existing models from becoming apparent. (More details are provided in the next section.) However, similar to the effects observed on downstream tasks such as MMLU and GLUE (Appendix B4), our experiments with hyperfitting only marginally impacted the models' summarization performance.
> > > >
> > > > ### **Additional updates - TopP Sampling:**
> > > > As became evident during our constructive discussion with reviewer p1Nv (see that thread for details), we realize that our longer sequence results are quite remarkable. Surprisingly, greedy decoding with hyperfitting outperforms Top-P sampling and becomes less repetitive as sequence length increases. This highlights the qualitative improvements that hyperfitting, counterintuitively, brings to the model itself.
> > > >
> > > > Although Top-P sampling was specifically introduced to mitigate repetitions, it does not address the underlying problem, as the predicted probabilities remain unchanged. Instead, it reduces the likelihood of selecting tokens that lead to repetitive patterns. However, as observed with TTR over long sequences, this risk persists and becomes increasingly likely as more tokens are generated.
> > > >
> > > > ### **Additional updates - Downstream Tasks & Instruct:**
> > > > As recommended by other reviewers, we further investigated hyperfitting’s effects on downstream tasks and instruction-tuned models. Appendix section B3 now includes preliminary experiments on hyperfitting already instruction-tuned models, and section B4 contains results on the MMLU benchmark. Across all these experiments, we find that the previously observed trends still hold: hyperfitting slightly decreases downstream performance but enhances long-sequence generation capabilities.
> > > >
> > > > ### **Conclusion:**
> > > > We thank you for your constructive suggestions and have addressed the requests regarding human annotations for larger models as well as the preliminary investigation of summarization tasks. Additionally, we have conducted more experiments on downstream tasks and further examined how hyperfitted models outperform Top-P sampling over longer sequence lengths.
> > > >
> > > > We continue to argue that hyperfitting is best understood as a peculiar phenomenon that opens up an intriguing avenue for future scientific exploration. Nonetheless, these applicable scenarios increase its appeal for future research.
> > > >
> > > > In light of these additions to the paper, would you consider revisiting and increasing your initial rating?

---

> > > > > ### Comment · Reviewer_KAD8 · 2024-12-02
> > > > >
> > > > > While I appreciate the authors' efforts in the rebuttal, its performance in other benchmarks is unclear at this stage. Thus I'l keep my original rating.

---

### Meta-Review · Area_Chair_CS3P · 2024-12-24

**Metareview:**

The paper introduces a finding denoted as the "hyperfitting phenomenon," which demonstrates that overfitting pre-trained Large Language Models (LLMs) on a small dataset can enhance their open-ended text generation capabilities. This approach mitigates the issue of repetitions in long-text generation. This is a counterintuitive phenomenon and it is explored and analyzed using various models and datasets, supported by a relatively large-scale human evaluation. More precisely, the study finds that hyperfitting leads to worse validation loss, deteriorating the language model's overall performance, but results in generated text that aligns better with human preferences and exhibits greater diversity according to established metrics. The authors propose some explanations for this phenomenon, including the sharpening of predictive distributions and improved top-rank prediction. Beyond text, preliminary experiments and qualitative evaluations are also conducted for image generation.

The reviewers acknowledge the novelty and significance of the phenomenon, as well as the experimental validation, including the extensive human evaluation. They raised questions regarding the practical implications, the explanations provided for the phenomenon, and the need for additional experiments with larger LLMs and more diverse datasets. During the rebuttal, the authors conducted experiments with larger LLMs, instruction-tuned models, and a new dataset (MMLU), addressing many of the concerns raised in the initial review. Following the rebuttal, the reviewers concluded that the novelty of the finding, along with the enhanced experimental contributions validating the observation across a range of models and tasks, represents a meaningful contribution. I recommend acceptance.

**Additional Comments On Reviewer Discussion:**

There were extensive discussions during the rebuttal, many of which focused on the impact of a paper that identifies a new phenomenon without clear practical applications. The authors were able to convince the reviewers of the significance of their findings.

---

### Decision · Program_Chairs · 2025-01-22

Accept (Poster)